# A five-century tree-ring record from Spain reveals recent intensification of western Mediterranean precipitation extremes

Marcos Marín-Martín[1], Ernesto Tejedor[1], Gerardo Benito[1], Miguel A. Saz[2], Mariano Barriendos[3], Edurne Martínez del Castillo[4], Jan Esper[4,5], Martín de Luis[2]

[1]Department of Geology, National Museum of Natural Sciences-Spanish National Research Council (MNCN-CSIC), 28006 Madrid, Spain
[2]Departamento de Geografía y Ordenación del Territorio. Instituto Universitario de Investigación en Ciencias Ambientales de Aragón. Universidad de Zaragoza
[3]Instituto de Diagnóstico Ambiental y Estudios del Agua (IDAEA-CSIC)
[4]Department of Geography. Johannes Gutemberg University, Mainz, Germany
[5]Czech Academy of Sciences, Global Change Research Institute, Brno, Czech Republic

*Correspondence to*: Marcos Marín-Martín (marcos.marin@mncn.csic.es)

**Abstract.** The Mediterranean basin, a recognized climate change hotspot, faces increasing hydroclimatic pressures, particularly from severe drought and precipitation events. To assess contemporary changes and potentially manage future impacts, it is crucial to understand the long-term context of this variability beyond the relatively short instrumental record. This study utilizes tree-ring records to reconstruct past hydroclimate in the Iberian Range of eastern Spain, a water-sensitive Mediterranean environment. We present a well-replicated tree-ring width chronology from *Pinus sylvestris* and *Pinus nigra* trees that calibrates and verifies significantly against cumulative instrumental precipitation over a 320-day period ending in June ($r = 0.749$; $p < 0.01$). The resulting 520-year reconstruction reveals substantial multi-centennial variability in precipitation and reveals an increase in the frequency and intensity of hydroclimatic extremes (both wet and dry) during the late 20th and early 21st centuries compared to the longer-term baseline. The reconstruction has a spatial representativeness centred over eastern and central Iberia and covaries with independent historical drought indices derived from rogation ceremony records during the late 18th and early 19th centuries. The documented intensification of hydroclimatic extremes is consistent with climate change projections and provides a baseline for evaluating ecosystem resilience and water resource vulnerability.

## 1 Introduction

The Mediterranean basin is widely recognized as a climate change hotspot, facing growing threats from hydroclimatic extremes such as prolonged droughts and periods of heavy rainfall (IPCC, 2021). These shifts threaten ecosystem stability, impacting even forest species that are generally well-adapted to dry conditions (Royo-Navascues et al., 2021, 2022; Vicente-Serrano et al., 2010). As pointed out by Lionello (2012), precipitation is a critical factor here, strongly influencing water availability for both nature and humans. How vulnerable conifers are to climate change, especially water shortages, is a well-studied topic

(Camarero et al., 1998, 2013; Linares and Tíscar, 2010; Navarro-Cerrillo et al., 2014; Vaganov et al., 2006). Therefore, the study of historical patterns of droughts and unusually wet periods is essential, not just academically, but for environmental sustainability and socioeconomic activities.

Researchers have already used tree rings in several important studies to piece together the Mediterranean's past hydroclimate. However, while valuable insights have been gained, the vast majority of these reconstructions have focused on drought indices, such as the Palmer Drought Severity Index (PDSI), the Standardized Precipitation Index (SPI), or the Standardized Precipitation-Evapotranspiration Index (SPEI) (Anchukaitis et al., 2024; Arsalani et al., 2021; Brewer et al., 2007; Esper et al., 2007, 2015; Fosu et al., 2022; Martin-Benito et al., 2013; Nicault et al., 2008; Royo-Navascues et al., 2022; Tejedor et al.,
2016, 2017; Touchan et al., 2005; Zhang et al., 2023). For instance, Nicault et al. (2008) created a major PDSI reconstruction spanning 1350-2000 CE across the basin, showing significant spatio-temporal variability in drought conditions. Building on such work, Tejedor et al. (2016) looked closely at drought patterns in Spain's Iberian Range by reconstructing the SPI, highlighting how the short instrumental period limits our understanding of long-term drought frequency and intensity. In contrast to the prevalence of index-based approaches, only a handful of studies have aimed to estimate actual rainfall amounts
for specific periods in the Mediterranean region (Akkemik et al., 2005; Griggs et al., 2007; Till and Guiot, 1990; Touchan et al., 2003, 2005, 2008, 2014), offering a different perspective on past hydroclimate.

Previous reconstructions have shown that precipitation in the Mediterranean has varied over the last centuries (Esper et al., 2015; Klippel et al., 2018; Tejedor et al., 2016). This includes major droughts that match up with historical records of crop failures and social problems (Dominguez-Castro and García-Herrera, 2016), but also times when the climate was relatively
stable with fewer extremes (Manrique and Fernandez-Cancio, 2000). This really underlines why understanding past climate changes is so important for preparing for—and managing—the impacts of future climate change on the region's water and forests.

This study draws on the long-term perspective of dendroclimatology to investigate potential shifts in the hydroclimatic regime of the Iberian Range. We focus on whether the second half of the 20th century and the early 21st century show a notable rise
in the frequency and intensity of extreme events, including both severe droughts and unusually wet years. This question becomes especially relevant when compared to studies based solely on the instrumental record. For example, Vicente-Serrano et al. (2025) analyzed data from 1871 to 2020 and observed marked hydroclimatic variability in the Mediterranean, which they primarily attributed to natural variability casting doubt on a strong anthropogenic signal during that timeframe. By extending the record over several centuries, our reconstruction offers a broader context in which to interpret these recent changes. This
expanded baseline helps assess whether current extremes are truly exceptional, contributing to the debate on natural versus human-driven influences, and highlighting possible links to shifts in regional atmospheric circulation (Hoerling et al., 2012).

Conversely, the extended temporal reach provided by the tree-ring record allows us to explore whether earlier centuries exhibited a different pattern of variability. This study explores whether the centuries preceding the mid 20th century were marked by comparatively longer intervals of relative hydroclimatic stability, with fewer dramatic swings between drought and periods of extreme precipitation. Establishing such a contrast would underscore the unusual nature of recent variability and align with paleoclimatic reconstructions that indicate periods of greater climatic consistency in Europe's past, prior to the pronounced warming and associated instability of the contemporary era (Büntgen et al., 2011, 2021; Serrano-Notivoli et al., 2023).

Finally, recent advances in the availability and quality of climate data products for the Iberian Peninsula provide a crucial opportunity to refine our understanding of past hydroclimate variability using tree rings. Specifically, the development of high-resolution gridded precipitation datasets offers a more robust basis for calibrating dendroclimatic models compared to what was available for some earlier studies. Previous reconstructions in the Mediterranean, while valuable, often explained a limited fraction of the instrumental climate variance (around 40 % in optimal cases) and frequently focused on drought indices rather than direct precipitation amounts. Consequently, leveraging these enhanced climate data resources alongside updated and extended tree-ring chronologies allows us to pursue several key objectives in this study: 1) To improve the explained variance of the hydroclimate reconstruction, thereby reducing uncertainty. 2) To extend the reconstruction further back in time by integrating new and existing tree-ring samples. 3) To focus on reconstructing quantitative precipitation, moving beyond drought indices to provide a more direct measure of past weather conditions. These objectives align with the need for more precise and longer-term hydroclimatic baselines in this climate change hotspot.

## 2 Materials and methods

### 2.1 Study area

This study was conducted in the province of Teruel, situated in the Aragón region of eastern Spain, specifically located on the Iberian Range. These mountains present a complex topography, with hills interspersed with plateaus and depressions (Peinado et al., 2017; Vergés and Fernàndez, 2006). Tree-ring samples were collected from five distinct locations within this range: Moscardón (MOS), Mosqueruela (LIN), Barranco de la Bellena (BEL), Javalambre (JAR) and Valdecuenca (VAN) (Figure 1A, 1D). These sites are located in relative proximity, all falling within a 100 km radius. The elevations of these sites vary, generally ranging from 1300 to 2000 meters above sea level, encompassing typical altitudes for the target pine species in the Iberian Range as documented in previous dendrochronological work (Tejedor et al., 2016).

The climate of the region is best classified as a continentalized Mediterranean type. It features pronounced seasonality with hot summers. However, deviating from classic Mediterranean patterns, a distinct dry season is not apparent; May is the month with the highest precipitation, followed by April. Consequently, the primary precipitation period extends from spring into early

summer, rather than being confined to winter. (Castro et al., 2005; Nicault et al., 2008). This pattern is illustrated by a climograph representing the study area's average conditions (Figure 1B), showing July and August as the months with the highest mean temperatures, with July also being the driest month. Spring and autumn typically exhibit precipitation peaks.

Although summers are generally dry, localized convective storm activity can occur, especially in the second half of August, which in some years can affect the latewood formation. Data for the climate diagram were extracted for the 1986–2015 period. Within this water-limited environment, precipitation variability acts as a primary limiting factor for forest productivity, a fundamental principle in dendroclimatology (Fritts, 1976), and studies have shown that Mediterranean pine forests respond strongly to moisture availability indices (De Luis et al., 2013; Pasho et al., 2011a; Royo-Navascues et al., 2022). The forests

at the study sites are dominated by Scots pine (*Pinus sylvestris* L.) (Figure 1C) and black pine (*Pinus nigra* Arn.), species valued in dendrochronology for their longevity and established sensitivity to climatic fluctuations, particularly moisture availability, as demonstrated in studies within the Iberian Range and other Mediterranean mountain environments (e.g. Camarero et al., 2013; Dorado-Liñán et al., 2019).

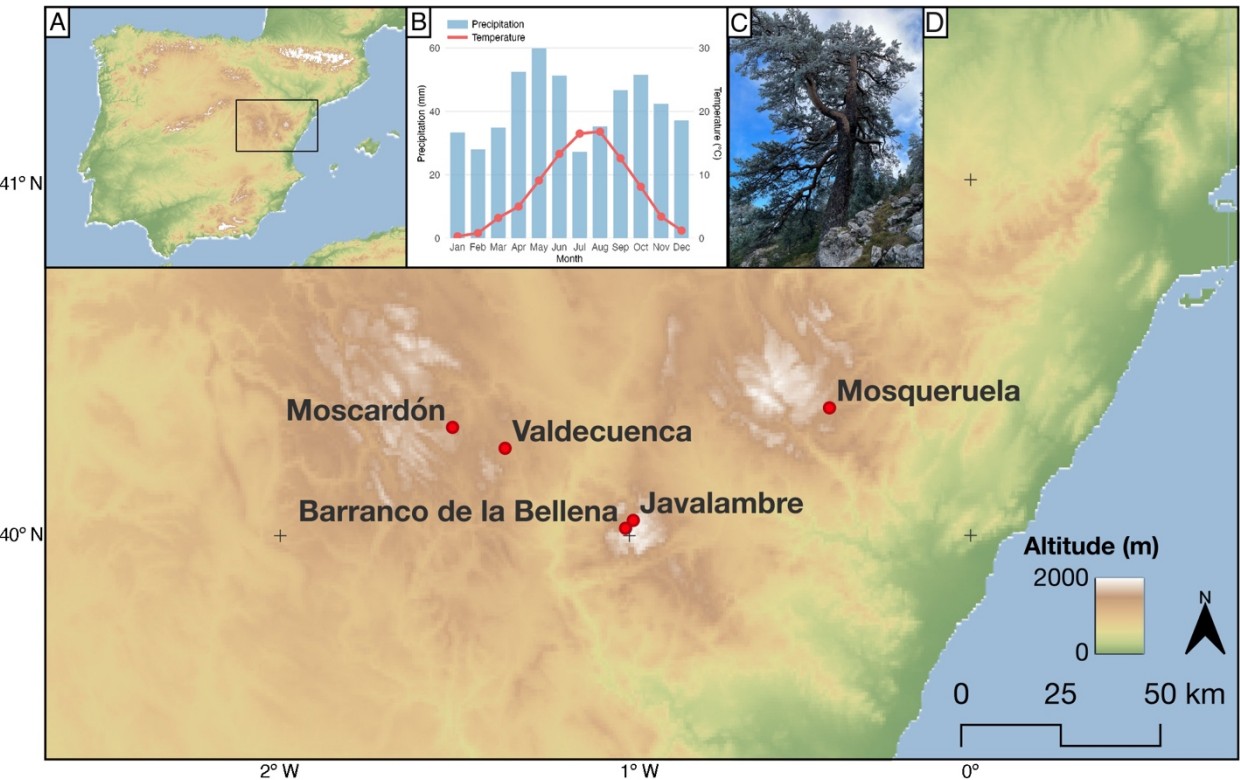

**Figure 1: A) Orographic map of the Iberian Peninsula and study area. B) Climate diagram showing monthly precipitation from the ROCIO 5 × 5 km grid (Peral García et al., 2017) and temperature from the Spain02 20 × 20 km grid (Herrera et al., 2016), both calculated as the average value of the 16 grid cells closest to the coordinate 40.3º N, 1.4º W, for the period 1986–2015. C) Photograph of a *Pinus sylvestris* at Barranco de la Bellena from which samples were taken. D) Study area with the five sampling locations pointed in red.**

## 2.2 Tree chronology development

The basis of this study is a tree-ring dataset comprising increment cores extracted from living *Pinus sylvestris* and *Pinus nigra* trees. Sampling was conducted in December 2024 across the five study locations. To extend the chronology further back in time and increase sample replication, these newly collected cores were integrated with existing tree-ring width measurement series obtained from previous sampling campaigns at the same sites conducted in 1994 and 2013 (Tejedor et al., 2016). Standard laboratory procedures for dendrochronological analysis were followed (Stokes and Smiley, 1968). Increment cores were allowed to air dry before being mounted on grooved wooden supports. The exposed transverse surface of each core was then carefully sanded using progressively finer grits of sandpaper until the cellular structure and individual ring boundaries were clearly visible under magnification. High-resolution (3200 dpi) digital images of the prepared cores were captured using a flatbed scanner. Individual core images were subsequently processed for measurement.

Annual ring widths were measured to the nearest 0.01 mm using the CooRecorder software (Larsson, 2005; Maxwell and Larsson, 2021). The process of cross-dating, which ensures that the correct calendar year is assigned to each annual ring, was performed by visually matching characteristic ring-width patterns among cores within the CDendro software. The accuracy of this visual cross-dating was statistically validated using the well-established COFECHA program (Holmes, 1983). COFECHA employs segmented time series correlation analysis to identify potential dating inconsistencies or areas of low agreement between individual series and the site master series, confirming the temporal integrity of the dataset.

After measuring the samples, each individual ring-width series was standardized to remove age/size-related trends and to minimize non-climatic noise. This was performed using the dplR package in R. Following an adaptive approach based on series length, a negative exponential curve or a cubic smoothing spline of variable stiffness was fitted to each series. Tree-ring indices were then calculated as ratios by dividing the raw ring-width measurements by the fitted curve values.

These individual indices were then combined into a regional chronology using a biweight robust mean, a method that minimizes the influence of outliers. Both a standard and a residual version of the chronology were computed. The residual chronology was selected for the final reconstruction because it is generated via prewhitening, a procedure that fits an autoregressive (AR) model to the standard chronology and removes the statistical autocorrelation inherent in tree growth. (Cook and Kairiukstis, 1990).

In addition, a sensitivity analysis was conducted to test the effect of an alternative method, variance stabilization, on our results. As detailed in the Supplementary Table S3, this approach did not improve the calibration skill. Therefore, the residual chronology was retained as it worked as the strongest predictor. After excluding some series with a low intercorrelation ($r < 0.3$) or that were inaccessible for remeasurement, the final dataset consisted of 173 individual tree-ring series from 103 different trees.

To justify the creation of a regional chronology, we assessed the coherence among the five sites by calculating inter-site correlations on their standardized residual series (Supplementary Table S1A and S1B). The results demonstrate a strongly cohesive network, with 9 out of 10 pairings showing positive and highly significant correlations ($p < 0.05$). The only pair that does not meet this strict threshold is Javalambre-Valdecuenca (JAR-VAN), with a correlation of $r = 0.19$ ($p = 0.057$). It is important to note that this pair also has the shortest overlapping period of all combinations, with only 97 years of common data (Supplementary Table S2). With fewer degrees of freedom ($df = 95$), a higher correlation coefficient is required to achieve statistical significance compared to pairs with centuries of overlap. Nevertheless, the JAR-VAN correlation is still positive and significant at a 90 % confidence level ($p < 0.1$).

Considering the overwhelming coherence of the rest of the network, this marginal result for the pair with the least data does not detract from the conclusion that a robust common climatic signal exists. Therefore, we deemed the combination of all five sites into a regional composite to be a valid and well-supported approach.

## 2.3 Climatic data

To relate tree growth to climate and select an appropriate target variable for reconstruction, nine different gridded precipitation datasets were obtained and evaluated, based on the hypothesis that higher resolution precipitation datasets might better capture the variability affecting tree-ring growth. The use of multiple climate datasets is crucial, as their representation of local climate can vary due to differences in station data density, interpolation algorithms and spatial resolution (Sun et al., 2018; Tapiador et al., 2012; Xiang et al., 2021), affecting the calibration of climate reconstructions. The datasets evaluated, covering a range of characteristics (see Table 1), included: SiCLIMA (Serrano-Notivoli et al., 2024), TerraClimate (Abatzoglou et al., 2018), ROCIO (Peral García et al., 2017), SPREAD (Serrano-Notivoli et al., 2017)), the European E-OBS dataset (v. 31.0e) (Cornes et al., 2018), ERA5-Land reanalysis (Hersbach et al., 2020) and the Climatic Research Unit Time Series (CRU TS v4) (Harris et al., 2020). For each dataset, monthly precipitation values were extracted for the grid points encompassing the study region and averaged to create regional climate series suitable for comparison with the tree-ring chronology.

**Table 1: Climate datasets used for calibration and verification.**

| Dataset | Spatial resolution | Temporal coverage | Temporal resolution | Developer |
|---------|--------------------|-------------------|---------------------|-----------|
| SiCLIMA | 0.005º | 1950–2020 | Daily | Universidad de Zaragoza (UNIZAR) |
| TerraClimate | 0.042º | 1958–2023 | Daily | Climatology Lab – UC Merced |
| ROCIO | 0.05º | 1951–2022 | Daily | Spanish State Meteorological Agency (AEMET) |
| SPREAD | 0.05º | 1950–2012 | Daily | Universidad de Zaragoza (UNIZAR) / CSIC |
| E-OBS | 0.1º | 1950–2023 | Daily | Royal Dutch Meteorological Institute (KNMI) |
| E-OBS | 0.25º | 1920–2023 | Daily | Royal Dutch Meteorological Institute (KNMI) |

| ERA5-Land | 0.28º | 1950–2024 | Daily | European Centre for Medium-Range Weather Forecasts |
|---|---|---|---|---|
| CRU TS | 0.5º | 1901–2023 | Monthly | Climate Research Unit – University of East Anglia |
| CRU TS | 1.0º | 1901–2023 | Monthly | Climate Research Unit – University of East Anglia |

## 2.4 Statistical analysis

All statistical computations and data processing were performed using the R language and environment for statistical computing (R Core Team, 2025). Dendrochronological analyses were primarily carried out using functions within the widely adopted dplR package (Bunn, 2008) and supplemented by the dendroTools package (Jevšenak and Levanič, 2018).

Chronology quality and signal strength were assessed using standard dendrochronological metrics, including the mean inter-series correlation (Rbar) (Briffa and Jones, 1990; Wigley et al., 1984), the signal-to-noise ratio (SNR) (Cook and Kairiukstis, 1990; Wigley et al., 1984) and the subsample signal strength (SSS) (Buras, 2017; Wigley et al., 1984).

To identify the most suitable climate dataset and the optimal temporal window for reconstruction, a two-stage correlation analysis using Pearson correlation coefficients was performed. First, based on previous findings suggesting strong correlations with approximately annual moisture integration (Tejedor et al., 2016), the regional tree-ring chronology was correlated with 12-month cumulative precipitation sums derived from each of the nine candidate climate datasets. This initial analysis systematically tested 12-month windows ending in each month of the calendar year (January to December) to select the climate dataset with the strongest overall relationship with tree growth.

Second, once the most promising dataset was identified from the monthly analysis, we employed the *daily_response* function within the dendroTools package (Jevšenak and Levanič, 2018) to precisely pinpoint the optimal climatic window at a finer temporal resolution. This function calculates Pearson correlations between the residual chronology and daily precipitation data aggregated over a wide range of moving window widths (lengths) and ending days, spanning the previous and current growth years. This systematic exploration of numerous, biologically relevant window combinations is designed to identify robust climate signals and minimize the risk of detecting potentially spurious correlations that can arise from single, arbitrary daily window tests (Torbenson et al., 2024). This detailed approach allows for a data-driven definition of the critical growth period beyond fixed monthly aggregations.

A linear transfer function model was selected to reconstruct the target precipitation variable identified through the correlation analyses, using the residual tree-ring chronology as the predictor. The model's predictive capability and temporal stability were assessed using a split-period calibration-verification scheme (Cook et al., 1999; Fritts, 1976; Meko, 1997). The instrumental climate record (1952–2022) was divided into two sub-periods (1952–1986 and 1987–2022). The model was calibrated on each sub-period and verified against the independent data of the other sub-period. Verification statistics, including the coefficient of determination ($r^2$) and the reduction of error (RE) statistic (Fritts, 1976; Nash and Sutcliffe, 1970), were calculated for both

periods. Following verification, the final transfer function was calibrated over the entire instrumental period (1952–2022) and applied to the tree-ring chronology over its reliable length (as determined by the SSS > 0.85 criterion) to generate the precipitation reconstruction.

To further assess the temporal stability of the transfer function, particularly potential changes in the regression parameters themselves, we implemented the bootstrapped transfer function stability test (BTFS) following Buras et al. (2017). This test evaluates the stability of the model's intercept, slope and explained variance ($r^2$), as well as the consistency of model significance over time. The instrumental period (1952–2022) was split into early (1952-1986) and late (1987–2022) sub-periods. Using 1000 bootstrap iterations, linear models relating the residual chronology to the target precipitation variable were fitted for resampled data within each subperiod. Ratios (late/early) of the resulting intercept, slope and $r^2$ were calculated for each iteration to generate empirical distributions. Stability was assessed by examining whether the 95 % confidence intervals of these parameter ratios contained the value 1 and whether the proportion of iterations where both sub-period models were simultaneously significant ($p < 0.05$) exceeded a threshold of 0.95. This provides a robust check against potential non-stationarity in the climate-growth relationship within the calibration period.

## 2.5 Rogation ceremonies

To provide an independent historical context for our tree-ring reconstruction, we utilized data on rogation ceremonies (*rogativas*), often prompted by environmental factors, from the dataset compiled by (Tejedor et al., 2019). These documented religious appeals primarily include *rogativas pro pluvia*, ceremonies asking for rain during perceived drought periods affecting agriculture and society, especially before widespread irrigation and modern forecasting (Barriendos, 1997; Martín-Vide and Barriendos, 1995). Records of *rogativas pro serenitate*, requesting the cessation of excessive rain or floods, were also qualitatively reviewed for context on wet extremes (Barriendos and Barriendos, 2021).

We adopted the established methodology (Barriendos, 1997; Tejedor et al., 2019) of converting the *pro pluvia* records into an annual semi-quantitative drought index (DI). This involved assigning ordinal intensity levels (0-3, with 0 indicating no relevant ceremony and 3 indicating the type of ceremony reflecting maximum perceived drought severity) based on ceremony type and aggregating these from the previous December to the current August for each location. For comparison with our reconstruction, we created a combined regional index by averaging the annual DI series from the "Ebro Valley" (DIEV) and "Mediterranean" (DIMED) clusters identified in Tejedor et al. (2019) representing lower-elevation agricultural areas relevant to our study region. This combined index covers the period 1650–1899 CE. The individual Teruel DI series was also examined separately due to its proximity to our tree-ring sites.

The core analysis involved calculating Spearman correlations between our August–June precipitation reconstruction and both the combined DIEV+DIMED index and the Teruel DI. Temporal stability was assessed using 50-year moving correlation

windows across the available 1650–1899 common period. This approach allows comparison while acknowledging the differing nature and inherent limitations of rogation-derived data versus tree-ring proxies.

## 3 Results

### 3.1 Tree-ring chronology

The final combined tree-ring dataset, comprising 173 series from *Pinus sylvestris* and *Pinus nigra* across the five study sites, spans the period 1505–2024 CE. The resulting regional residual chronology exhibits robust statistical properties indicative of a strong common signal suitable for climate reconstruction. The average length (age) of the individual tree-ring records included in the chronology was 182 years, with the shortest series spanning 31 years and the longest spanning 576 years. Following standardization procedures, the temporal structure of the final residual chronology (res) was evaluated. The first-order autocorrelation (AR1) coefficient for this residual chronology was 0.039. This low value indicates that standardization effectively removed the non-climatic, biological persistence inherent in tree growth. The instrumental precipitation target also exhibited negligible first-order autocorrelation (AR1 = –0.037). This ensures that both predictor and predictand are suitable for correlation analysis. More critically, the residuals of the final linear reconstruction model were tested for autocorrelation using the Durbin-Watson statistic; the test was not significant (DW = 2.14; p = 0.729), confirming that the model meets the assumption of independent errors required for linear regression (Cook and Kairiukstis, 1990). To assess the high-frequency variability inherent in the raw ring-width measurements, the mean sensitivity (MS) was calculated across all series, resulting in an average value of 0.304. This metric quantifies the relative change in ring width from one year to the next, with higher values generally indicating a stronger year-to-year growth response often associated with sensitivity to limiting climatic factors (Fritts, 1976). MS values above 0.3 represent sensitive measurement values (Grissino-Mayer, 2001).

The mean inter-series correlation (Rbar), calculated over the common period, was 0.273, signifying a substantial degree of shared variance among the individual tree-ring series (Briffa and Jones, 1990; Wigley et al., 1984). Furthermore, the signal-to-noise ratio (SNR) was 64.936, which shows that the common signal variance strongly outweighs the noise variance specific to individual series (Cook and Kairiukstis, 1990; Wigley et al., 1984).

The temporal evolution of the chronology's signal strength and replication is presented in Figure 2. This graphic displays the standardized residual chronology (blue line), the subsample signal strength (SSS, red line), the 51-year running Rbar (yellow) and the number of samples per year (sample depth, green line). The SSS statistic quantifies the retention of the common signal as sample depth decreases back in time (Buras et al., 2017; Wigley et al., 1984). Based on the commonly applied threshold of SSS > 0.85 (Buras et al., 2017; Cook and Kairiukstis, 1990), the chronology is considered reliable for climate reconstruction from 1505 through to 2024. This reliable portion of the chronology served as the predictor variable for the subsequent precipitation reconstruction.

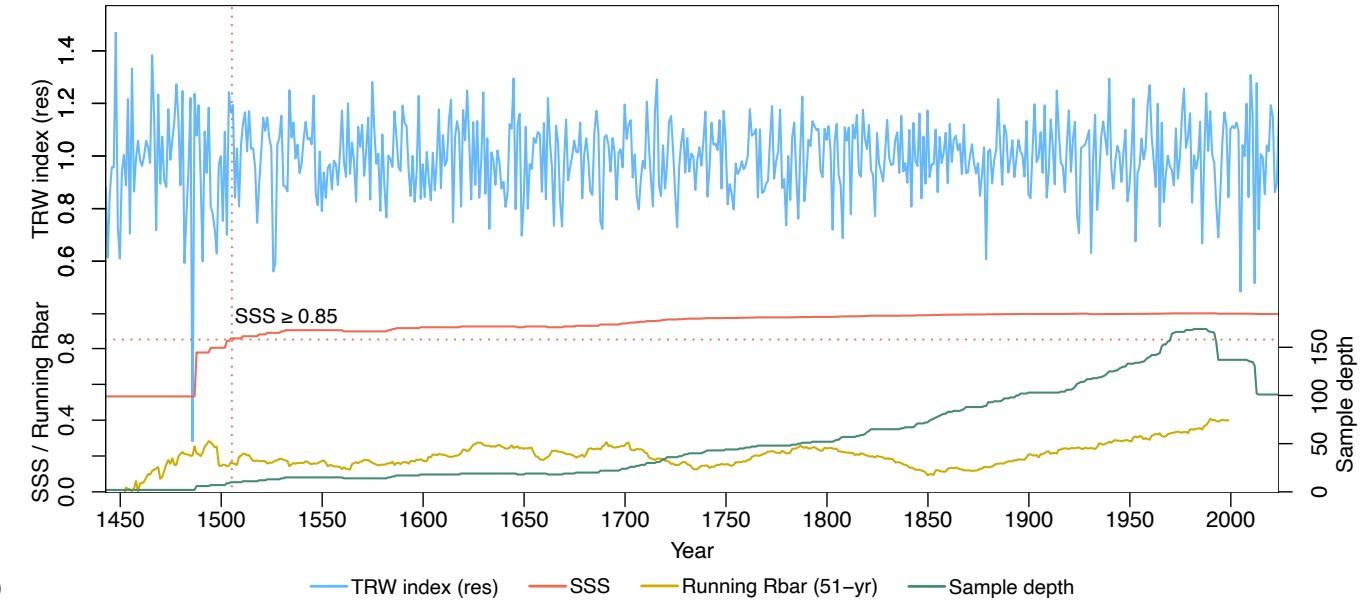

**Figure 2: Residual chronology (*blue*), SSS (*red*), running 51-year Rbar (*yellow*) and number of cores per year and SSS (*green*).**

Additionally, to ensure the robustness of our findings to the chosen methodology, a comprehensive sensitivity analysis was performed. An alternative reconstruction was developed using a variance stabilization method (chron.stabilized in dplR; Frank et al., 2007; Wigley et al., 1984), which corrects for variance changes associated with fluctuating sample replication over time.

A thorough statistical comparison revealed that both reconstructions are highly similar (Pearson r = 0.92) and exhibit no significant differences in their means (paired t-test, p = 0.087) or their overall probability distributions (Kolmogorov-Smirnov test, p = 0.246). Crucially, a specific analysis of extreme events showed no statistically significant differences in their frequency or temporal distribution (McNemar's and Chi-squared tests, p > 0.48). Given that the study's conclusions are robust to the choice of method, and considering the theoretical suitability of the residual chronology for calibration, the latter was retained

for all analyses presented. Full details of this comparative analysis can be found in the Supplementary material (Table S3).

**3.2 Climatic datasets comparison**

The comparative analysis correlating the regional tree-ring chronology with 12-month cumulative precipitation sums from the nine candidate climate datasets revealed both consistent seasonal patterns and significant differences in correlation strength between datasets. Figure 3 visualizes these Pearson correlation coefficients, mapping the correlation strength for each dataset

(y-axis) against 12-month precipitation windows ending in each calendar month (x-axis), for the common period covered by the nine databases (1958–2012).

Across most datasets, the strongest positive correlations consistently occurred for precipitation periods ending in the summer months (June, July, August). This indicates that tree growth in the study region primarily integrates moisture availability

accumulated during the preceding year leading up to and including the current year growing season. Notable differences emerged based on dataset characteristics; the high-resolution Spanish datasets (ROCIO, SPREAD) and the E-OBS dataset generally yielded stronger correlations compared to the global reanalysis (ERA5-Land) and the coarser CRU TS products.

Specifically examining the peak response period, the highest correlations were observed for 12-month windows ending in the summer months, with the ROCIO, SiCLIMA and SPREAD datasets showing strong relationships ($r \approx 0.70$). Due to its high correlation strength during this optimal period and its extended temporal coverage (expanding 1951–2022, see Table 1), the ROCIO dataset was selected as the most suitable climate product for the subsequent detailed daily response analysis aimed at defining the precise target variable for reconstruction.

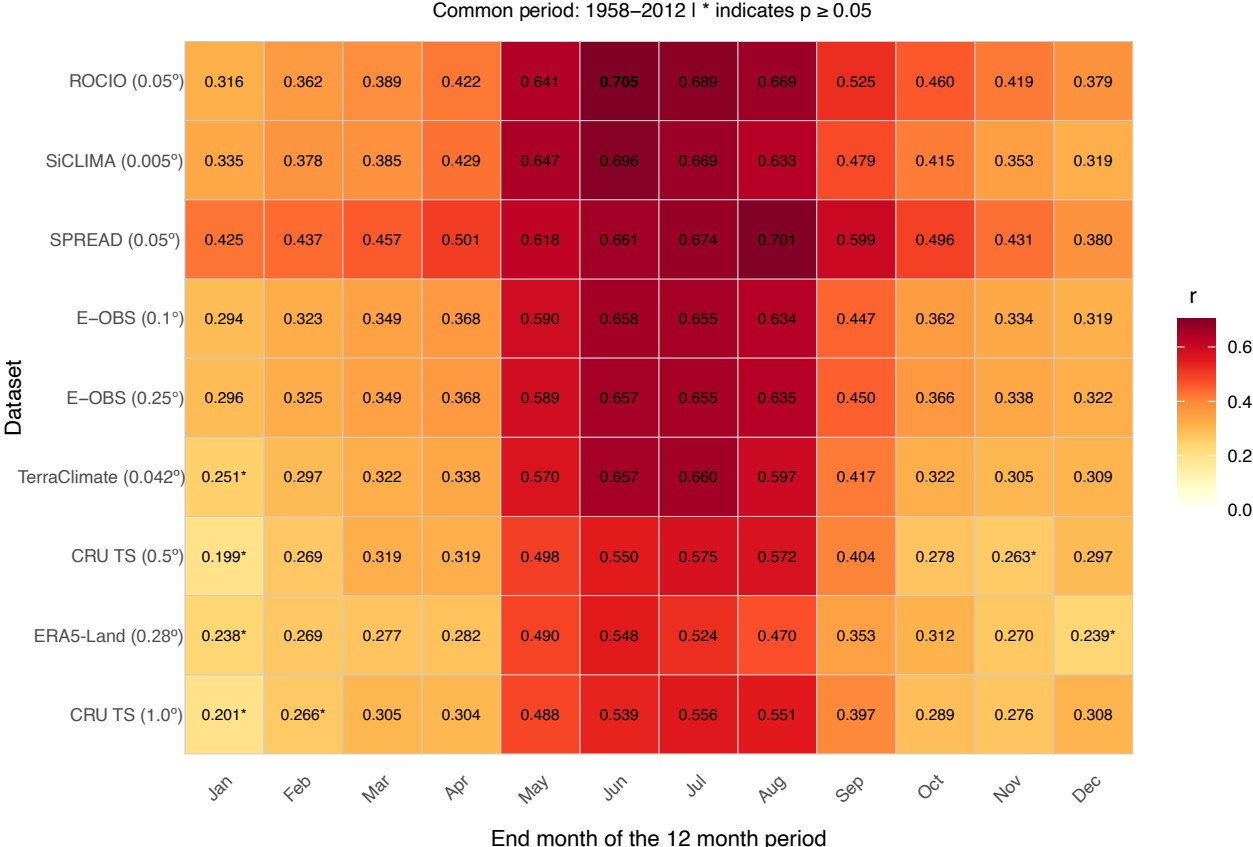

**Figure 3: Pearson correlation heatmap between the residual tree-ring chronology and accumulated period over 12-month periods for eight instrumental climate datasets (1958–2012). Insignificant correlations (p > 0.05) are marked with an asterisk (\*).**

To further validate our hypothesis that higher-resolution precipitation datasets display superior reconstruction skill when compared to lower-resolution options, we performed a calibration and verification exercise for our tree-ring based precipitation reconstruction using two distinct instrumental datasets of differing spatial resolutions (Figure 4). This analysis focused on the

common period of 1958–2012, selected to ensure consistency across all databases utilized in this study. For this comparative
calibration, we aggregated precipitation into an 11-month window spanning August of the previous year to June of the current
year. This specific window was necessary because the CRU TS dataset, one of the calibration targets, only provides monthly
resolution, which contrasts with the daily data from the ROCIO dataset ultimately used to refine our final reconstruction. When
calibrated against the widely-used CRU TS gridded dataset (at 0.5° resolution), the reconstruction achieved a Pearson
correlation coefficient (r) of 0.54 and a root mean square error (RMSE) of 98.39 mm. In contrast, calibration against the
national high-resolution ROCIO dataset from AEMET demonstrated a notable improvement in reconstruction skill, yielding a
Pearson r of 0.71 and a reduced RMSE of 80.99 mm. These results highlight the enhanced performance achieved when using
the higher-resolution, locally sourced ROCIO dataset for calibration.

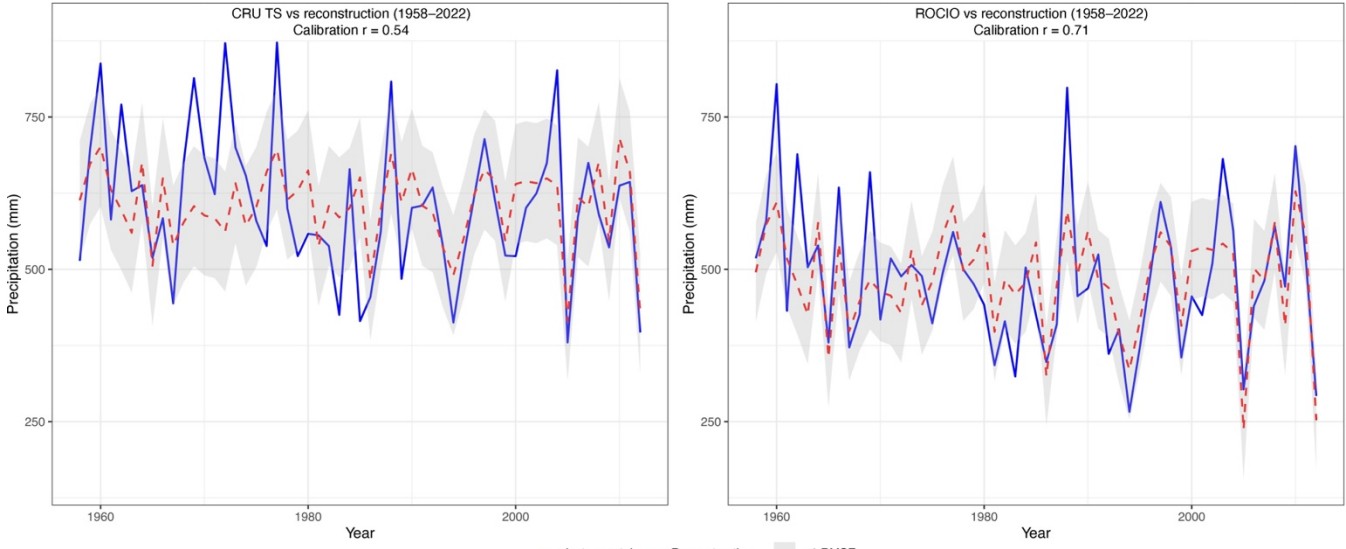

**Figure 4: Calibration of the pine residual chronology against CRU TS and ROCIO previous-year August to current-year June
precipitation sums (blue curves) from 1958–2012. The red dashed line represents the chronology-based precipitation reconstruction.
Grey shades represent ±1 RMSE.**

## 3.3 Precipitation reconstruction

### 3.3.1 Target variable identification

After the selection of the ROCIO dataset, a detailed *daily_response* function analysis was performed using
the dendroTools package (Jevšenak and Levanič, 2018) to pinpoint the optimal climate window driving tree growth. This
involved correlating the residual chronology against daily ROCIO precipitation aggregated over numerous moving windows
varying in width and ending day across the previous and current growth years. The results, visualized in Figure 5, map the
correlation strength across this parameter space.

The systematic exploration revealed the strongest positive correlation (r = 0.749) occurred for a window width of 324 days, starting on August 11th of the previous year and ending on July 1st of the current growth year. However, considering practical interpretation and alignment with recognized seasonal climate phases in the Mediterranean, a slightly adjusted window of 320 days was selected. This window, spanning August 16th of the previous year to June 30th of the current growth year, maintains a very high correlation (r = 0.748), negligibly different from the mathematical maximum.

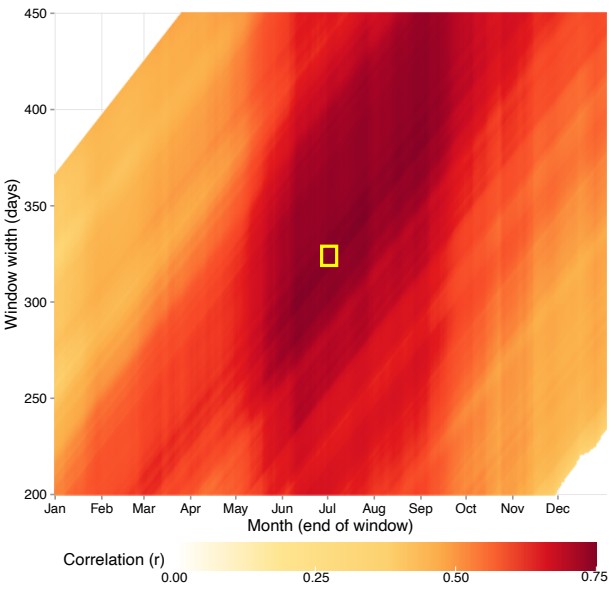

**Figure 5: Daily climate-growth response analysis showing Pearson correlation coefficients between the residual tree-ring chronology and daily accumulated precipitation. The heatmap explores a vast parameter space to identify the optimal climate window influencing tree growth. The y-axis represents the width of the precipitation window in days (from 200 to 450 days). The x-axis represents the ending day of this window, spanning from January to December of the current growth year. The colour of each pixel indicates the strength of the correlation (r). Only statistically significant correlations (p < 0.05) are shown. The yellow box highlights the pixel with the maximum positive correlation (r = 0.749), which corresponds to a window of 324 days ending on July 1st.**

### 3.3.2 Reconstruction model development

A simple linear regression of the target predictand on the site chronology was developed using the selected 320-day cumulative precipitation (prior August 16–current June 30) as the predictand and the regional residual tree-ring chronology as the predictor. The model's performance and temporal stability were evaluated using a standard split-period calibration-verification procedure over the instrumental period (1952–2022), divided into early (1952–1986) and late (1987–2022) subperiods (Cook et al., 1999; Meko, 1997).

The results of this validation are shown in Figure 6. The model demonstrated considerable skill in capturing the observed precipitation variability. During the early calibration period (1952–1986), the model explained 47 % of the variance (r = 0.689), while during the late calibration period (1987–2022), it explained 63 % of the variance (r = 0.797). Additionally, the model showed robust predictive skill when tested against independent data in the verification steps. The reduction of error (RE) statistic, which assesses predictive skill relative to simply using the calibration period mean (Fritts, 1976), yielded positive values for both periods. The RE for the early verification period (predicting 1952–1986 based on the late calibration) was 0.474 and the RE for the late verification period (predicting 1987–2022 based on the early calibration) was 0.634. These positive RE values confirm that the model possesses genuine predictive ability beyond the calibration data (Cook et al., 1999). Regarding the bootstrapped transfer function stability (BTFS) test presented by Buras et al. (2017), all intercept, slope and $r^2$ were considered stable with our data at a 95 % confidence interval.

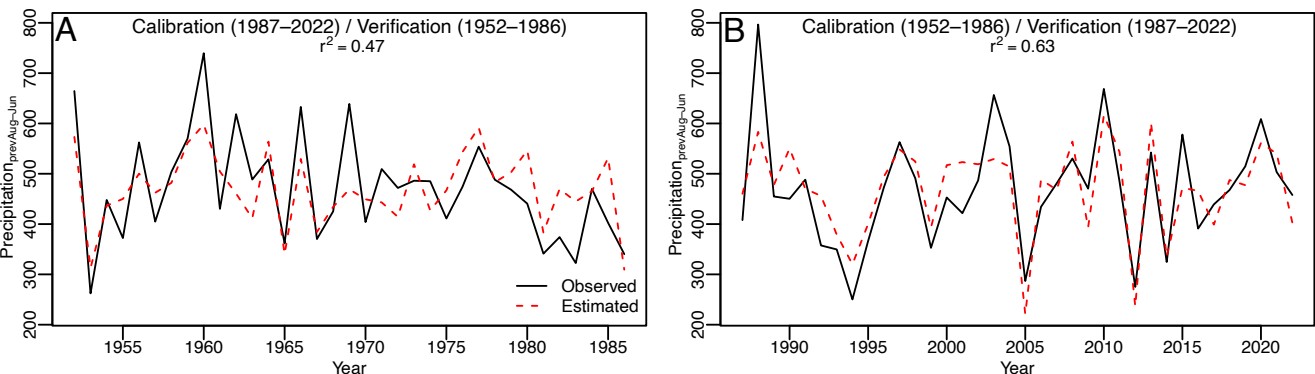

**Figure 6: Calibration and verification results of the 11-month precipitation reconstruction ending in current-year June 30 ($\alpha = 0.05$). A) Calibration period: 1987–2022, verification period: 1952–1986. B) Calibration period: 1952–1986, verification period: 1987–2022.**

Given the successful calibration and verification results, the final linear transfer function was calibrated using the entire instrumental period (1952–2022). This final model was then applied to the regional residual chronology over its reliable length determined by the SSS > 0.85 criterion (1505–2024 CE), yielding the final reconstruction of prior August 16–current June 30 cumulative precipitation. The final linear regression model, calibrated over the full instrumental period (1952–2022), established the quantitative relationship between the regional residual tree-ring chronology index (res) and the target cumulative precipitation in mm. The resulting equation used for the reconstruction is displayed in Eq (1):

$$mm = 0.5581 \cdot res + 207.25 \tag{1}$$

The final reconstructed precipitation time series, spanning the reliable period from 1505 to 2024 CE (determined by SSS > 0.85), is presented alongside several diagnostic elements in Figure 7. The root mean square error (RMSE), calculated during the calibration period (1952–2022), quantifies the typical magnitude of the residuals (differences) between the observed instrumental precipitation and the model's prediction based on tree rings (Fosu et al., 2022). It thus provides a standard measure

of the reconstruction's inherent uncertainty or prediction error. This uncertainty range (mean ±1 RMSE) is visualized on the plot by the dashed blue lines. Additionally, an 11-year moving average (green line) is overlaid on the annual reconstruction to smooth high-frequency (year-to-year) fluctuations and emphasize lower-frequency (decadal and multi-decadal) variability,

which aids in identifying prolonged wet or dry periods (Arsalani et al., 2021; Fosu et al., 2022). Finally, to objectively identify individual years characterized by exceptional hydroclimatic conditions relative to the long-term baseline, we utilized percentile thresholds derived from the full reconstructed series (1505–2024). This non-parametric approach defines extremes based on their rank within the data distribution. Specifically, years falling below the 1st percentile were classified as extremely dry (dark red squares) and those between the 1st and 5th percentiles as severely dry (light red squares). At the wet end, years exceeding

the 99th percentile were flagged as extremely wet (dark blue squares) and those between the 95th and 99th percentiles as severely wet (light blue squares) (Figure 7). The use of percentile thresholds provides a robust method for identifying extreme events (Zhang et al., 2011). These visual aids collectively facilitate the interpretation of the reconstructed precipitation history, highlighting its uncertainty, dominant timescales of variation and the occurrence of past extremes (Manrique and Fernandez-Cancio, 2000). Visual inspection of Figure 7 highlights periods characterized by distinct wet or dry conditions, as well as shifts

in variability. For instance, the reconstruction identifies notable drought periods, with years like 1526, 1527, 1879, 1931, 2005 and 2012 falling below the 5th percentile, and exceptionally wet periods, with years such as 1534, 1546, 1575, 1645, 1716, 1940, 2010 and 2013 exceeding the 95th percentile. To further investigate the temporal dynamics of hydroclimatic variability, we calculated a 51-year running standard deviation of the reconstructed precipitation series. This metric, which quantifies the magnitude of mid to high frequency fluctuations (Von Storch and Swiers, 1999), is displayed in Fig. 7 (green dashed line,

right-hand axis). The running SD reveals a period of pronounced climatic stability with low variability centred around the mid-19th century. This contrasts sharply with a subsequent, persistent rise in volatility that begins around the start of the 20th century and accelerates markedly after 1975, reaching values that are unprecedented in the context of our 520-year reconstruction.

To provide robust quantitative evidence for this observed intensification, we analyzed the frequency of extreme events per

370 century (Table 2). The analysis reveals a high contrast between the hydroclimatically stable 19th century, which recorded only nine events outside the P10–P90 range, and the recent period. The 21st century, though spanning only 24 years, has already accumulated eight such events. This intensification is most pronounced for the rarest occurrences: the 2001–2024 period has registered four exceptionally rare events (two <P01 and two >P99). This corresponds to an occurrence rate of 16.7 %, an order of magnitude higher than the 1.6 % average rate for such events across the preceding five centuries (1505–2000). These

375 numerical results offer strong statistical support for the visual evidence from the running SD and confirm the recent and anomalous intensification of hydroclimatic extremes in our study region.

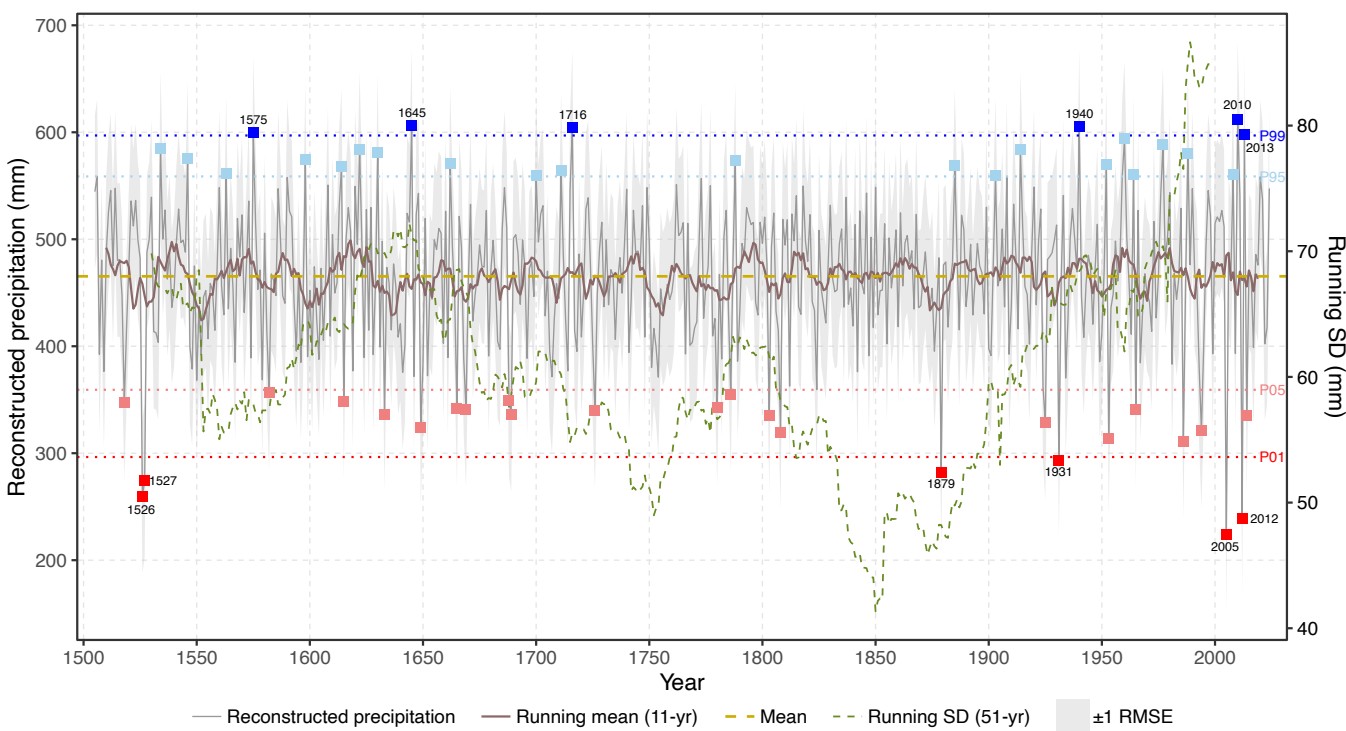

**Figure 7: Annual rainfall reconstruction between August 16 of the previous year and June 30 of the current year back to 1505 (grey curve). A 11-year rolling mean is shown in brown. Dark blue squares represent years above 99th percentile, light blue squares between 95th and 99th percentiles, dark red below 1st percentile and light red between 1st and 5th. The variability in the reconstructed values is shown with a green dashed line showing a 51-year running standard deviation.**

**Table 2: Frequency of extreme events per century, with breakdown by severity.**

| Period (CE) | Total years | Dry years (<P10) | <P05 | <P01 | Wet years (>P90) | >P95 | >P99 | Total extremes |
|---|---|---|---|---|---|---|---|---|
| 1505–1600 | 96 | 10 | 4 | 2 | 12 | 5 | 1 | 22.9 % |
| 1601–1700 | 100 | 13 | 7 | 0 | 9 | 6 | 1 | 22.0 % |
| 1701–1800 | 100 | 8 | 3 | 0 | 9 | 3 | 1 | 17.0 % |
| 1801–1900 | 100 | 6 | 3 | 1 | 3 | 1 | 0 | 9.0 % |
| 1901–2000 | 100 | 12 | 6 | 1 | 14 | 8 | 1 | 26.0 % |
| 2001–2024 | 24 | 3 | 3 | 2 | 5 | 3 | 2 | 33.3 % |

### 3.3.3 Spatial representativeness

The spatial pattern of correlation between the reconstructed precipitation series (prior August 16–current June 30) and the corresponding gridded instrumental precipitation from the E-OBS 0.1° dataset (Cornes et al., 2018) over the 1951–2023 period is presented in Figure 8. Correlations shown are statistically significant (p < 0.05). E-OBS was used due to its spatial extent, which encompasses all Europe and the whole Mediterranean basin, as well as because of its high correlation with the residual chronology in the study area (r = 0.68).

The map indicates that the highest positive correlation coefficients are concentrated over western Mediterranean, particularly
across eastern and central Iberia, which includes the study region where the tree-ring data originate. Positive correlations also extend significantly into southern France and northern Italy. Weaker, more fragmented areas of positive correlation are apparent in parts of northern Morocco. Regions outside of these specified areas generally show low or non-significant correlations with the reconstruction.

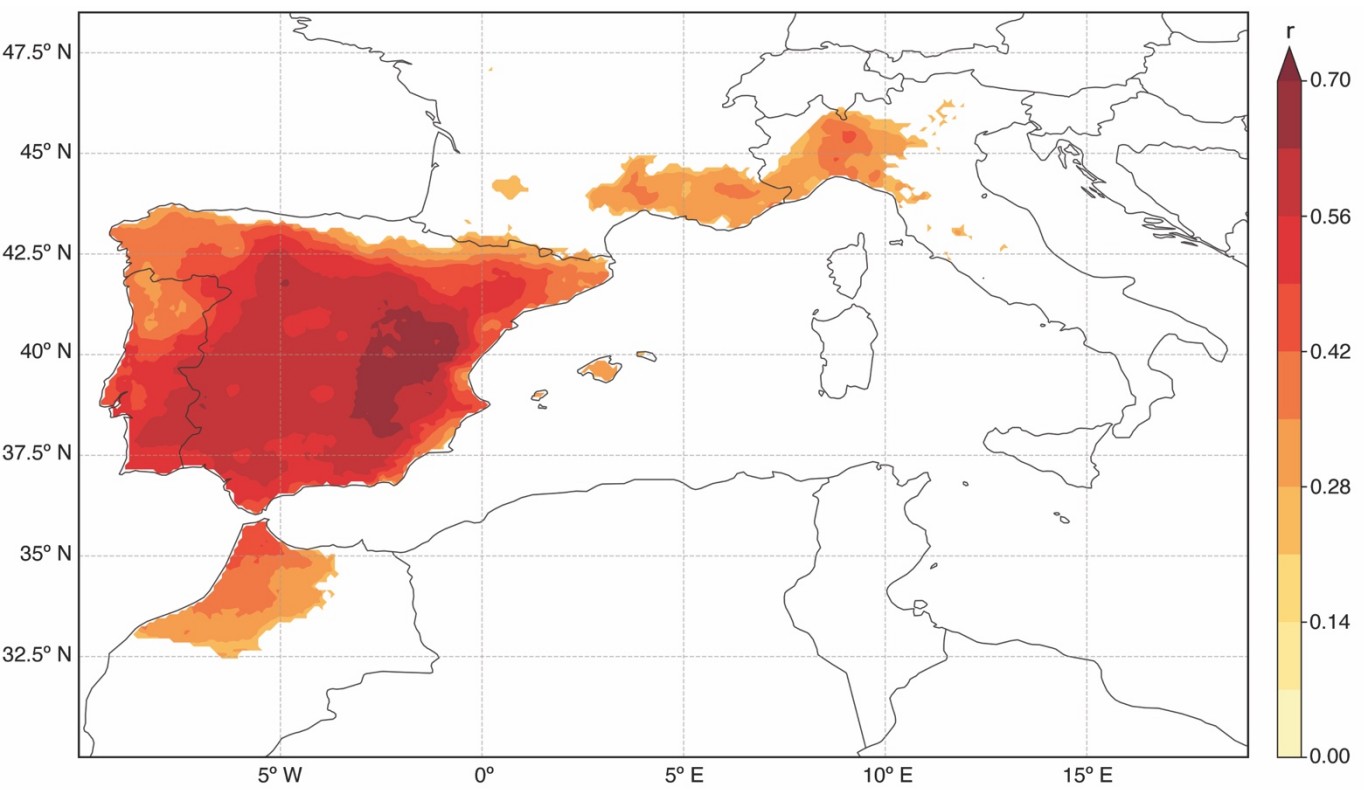

**Figure 8: Spatial correlation patterns between the tree-ring residual chronology and accumulated rainfall during 320 days prior to July 1st (1951–2023).**

### 3.3.4 Rogation-derived drought indices

We compared our tree-ring based reconstruction of previous August–current June precipitation (1505–2024) with independent historical drought indices derived from rogation ceremony records (1650–1899 CE), sourced and processed following Tejedor
et al. (2019). This comparison aims to identify potential shared hydroclimatic signals, while explicitly acknowledging the inherent limitations and differing nature of these two proxy types.

It is crucial to recognize that rogation records, while valuable historical proxies, differ fundamentally from tree-ring data. They represent a societal response primarily triggered by perceived drought impacts on agriculture, particularly cereal harvests, although influenced by the broader context of economic conditions and religious beliefs (Barriendos, 2005; Brázdil et al.,

2005). While historical archives can present challenges like gaps or inconsistencies, careful selection of sources, often administrative records known for better continuity, helps mitigate these issues. Rogations typically reflect the cumulative impact of water deficit leading to agricultural stress, often accumulating over winter and spring, though responses could become continuous during exceptionally severe droughts. Furthermore, the frequency and perhaps reliability of rogations demonstrably decreased from the mid-19th century onwards (Tejedor et al., 2019). This decline was linked to societal shifts

like growing anti-clerical sentiment and improved market access in developing areas, leading to uneven abandonment of the practice across regions and complicating comparisons in later periods.

Despite these significant caveats, the comparison revealed periods of statistically significant negative correlation between our reconstruction and the rogation-derived indices, as expected (lower reconstructed precipitation/higher drought stress should correlate with a higher rogation DI). When correlating our reconstruction with the combined DIEV+DIMED index

(representing lower-elevation Ebro Valley and Mediterranean coastal areas), we found a consistent period of significant ($p < 0.1$) negative correlation in 50-year moving windows centred roughly between the late 1790s and early 1800s (covering analysis windows from approx. 1769–1818 to 1780–1829). The Spearman correlation coefficients during this interval were modest, typically ranging between $r = –0.29$ and $r = –0.35$. While not indicating a strong predictive link, the persistence of this significant, albeit weak, signal over several decades suggests that both the trees and the lowland agricultural communities were

responding to shared hydroclimatic variability during this dynamic period encompassing the end of the 18th and beginning of the 19th century.

The comparison with the individual Teruel DI, the closest rogation site to our study area, yielded more varied results. Extremely high negative correlations ($r = –0.87$) were found for 50-year windows starting between 1694 and 1709. However, these calculations are based on a very small number of overlapping data points ($n = 5$) within those specific early windows in the

Teruel record and must be interpreted with extreme caution. Although this low sample size prevents drawing statistically robust conclusions from our comparison alone, it is noteworthy that this period falls within the Late Maunder Minimum (approx. 1675–1715), an interval documented through historical sources, including rogation records from nearby regions like Murcia, as being characterized by frequent and severe drought conditions in parts of the Iberian Peninsula (Machado et al., 2011; Rodrigo and Barriendos, 2008). More reliably, significant negative correlations reappeared in windows starting from 1787

through to 1830. Within this timeframe, correlations ranged from approximately $r = –0.37$ to $r = –0.56$, with sample sizes increasing to more acceptable levels ($n = 14$ to $n = 25$). This suggests a more dependable link between drought perceived in Teruel and regional tree growth during the late 18th and first half of the 19th century. This suggests a more reliable link between drought perceived in Teruel and regional tree growth during the late 18th and first half of the 19th century. A summary of the moving correlation analysis is presented in Table 3.

**Table 3: Main periods of significant correlation between reconstructed precipitation and regional rogation-derived drought indices. The table highlights the core intervals where a statistically significant negative correlation (Spearman r, $p < 0.05$) was identified between the reconstructed precipitation and the two main regional rogation clusters, based on a 50-year moving window analysis.**

**For each regional index, the table displays the period of significance, the average Spearman coefficient across that period and a measure of data density, expressed as the number of years with recorded rogation ceremonies relative to the total duration of the interval.**

| Area | Period (CE) | Mean Spearman correlation | Number of years with rogation ceremonies within the period |
|---|---|---|---|
| Mediterranean Coast (DIMED) | 1738–1801 | –0.32 | 58/64 |
| Ebro Valley (DIEV) | 1747–1808 | –0.35 | 56/62 |

Further validation involved comparing the nineteen wettest years (above the 95 % percentile before 1950) in our August-June precipitation reconstruction with qualitative information from historical documents describing extreme weather events, focusing on evidence of flooding or persistent rainfall occurring between the preceding autumn and June of the reconstructed year. This timeframe is consistent with the period integrated by the tree-ring data. Despite the inherent differences between a regional, integrated reconstruction and specific historical event documentation, fifteen positive correspondences were found. Indeed, most of the wet years are associated with multiple flood records in the Ebro basin or the Mediterranean rivers.

For instance, the reconstructed wet year of 1645 is consistent with historical reports of river flooding in the Ebro River (mid-February) and in the Onyar River (April), as well as excessive rainfall and very cold weather from January to April in northeastern Spain, which affected harvests. Similarly, in 1662 there were reports of both heavy rainfall in the previous autumn, which affected several Mediterranean streams persistent rainfall from March to May. In contrast, 1575 is only partially consistent with documentary reports, with rains documented in April, but with a widespread, albeit mild, drought in central and eastern Spain.

Other wetter years occurred during the highly variable "Maldà anomaly" (1760–1800; Barriendos and Llasat (2003)), which records mixed historical signals with both widespread rainfall and drought years. The wet year of 1777 is supported by documented heavy rains that occurred in October 1776, causing catastrophic flooding in the Túria River (Ruiz et al., 2014) and the Rambla del Poio (the same area affected by the 2024 flood in València), followed by multiple winter floods (January to March) in southern and eastern Spain. The wet year of 1788 was preceded by continuous heavy rainfall in October 1787, which caused the largest historical flood in the Ebro River (Balasch et al., 2019) followed by a wet winter and spring with flooding in western Iberian rivers.

In the late 19[th] century, the wet year of 1885 shows a good agreement with documentary evidence of multiple floods in Mediterranean and Ebro basin rivers starting in the autumn season of 1884 (September and November) and continuing with the winter to late spring floods (Balasch et al., 2019; Machado et al., 2011). Other significant wet years in our record with similar flooding patterns include 1888 and 1889, both of which report flooding from autumn to late spring in more than 20 rivers in the Ebro and Mediterranean basins.

While fundamental differences exist between continuous biological proxies and curated documentary series based on societal responses, the periods of agreement suggest both are capturing hydroclimatic fluctuations severe enough to impact ecosystems and human activities. The lack of a consistently strong correlation throughout the entire record likely arises from a combination of factors inherent to the proxies. The historical archive itself can be discontinuous due to issues like lost or damaged documents, which naturally complicates robust statistical comparisons over long timescales. Furthermore, our reconstruction represents an annually integrated, 320-day signal reflecting a physiological response to moisture, whereas rogation records are event-triggered, non-standardized societal responses to perceived agricultural stress that may not always align in their timing or seasonal emphasis.

## 4 Discussion

This study presents a novel, multi-centennial reconstruction of annual precipitation for the Iberian Range in eastern Spain, spanning the period 1505–2024 CE. Derived from a robust network of *Pinus sylvestris* and *Pinus nigra* tree-ring width series, the reconstruction targets the 320-day period from August 16th of the previous year to June 30th of the current growth year. Physiologically, this window was strategically selected through detailed response function analysis to capture the critical moisture accumulation phase influencing annual growth. This extended period is crucial because autumn and winter precipitation, often accumulating as snowpack at these mountain sites, recharges soil moisture during tree dormancy. The release of this water during spring snowmelt then directly supports earlywood formation (Pasho et al., 2011a, b), while the window still excludes the peak summer drought period (July to mid-August) where growth is typically limited by intense water stress (Camarero et al., 2013).

A relevant question concerns the choice of the reconstruction target, specifically why precipitation was selected over a more integrated drought index like the Standardized Precipitation-Evapotranspiration Index (SPEI) (Beguería and Vicente-Serrano, 2011). To address this, we conducted a direct comparative analysis to empirically determine the optimal target variable. We performed a series of calibrations testing both our residual and standard chronologies against three potential climate targets: our 320-day precipitation window, a standard 12-month precipitation window, and a 12-month SPEI.

The results, detailed in the Supplementary material (Table S4), demonstrate a clear and consistent hierarchy in model performance. Our selected 320-day window produced a stronger calibration than a standard 12-month window. Crucially, the model with the highest predictive skill was the one used in our study, which calibrates the residual chronology against 320-day precipitation Therefore, the selection of this specific target and predictor is not only based on our initial response function analysis but is also validated by this comprehensive comparative test as the statistically most robust approach for this dataset. It is worth noting, however, that while the 320-day window is empirically superior, the calibration strength of the 12-month window remains high, with only minor differences between the two.

The linear transfer function model developed using the ROCIO precipitation dataset exhibited significant skill and temporal stability, validated through rigorous split-period calibration-verification and BTFS tests (Buras et al., 2017) confirming its suitability for reconstructing past precipitation. The selection of ROCIO, a high-resolution gridded dataset from observed records, was informed by our comparative analysis which generally showed that datasets with higher spatial resolution yielded stronger correlations with the tree-ring chronology compared to coarser products (e.g., CRU TS, ERA5-Land). The robustness of the reconstruction is further demonstrated by the coherent signal and strong calibration statistics achieved when utilizing multiple high-resolution datasets. Our findings strongly advocate for using high-resolution precipitation datasets whenever available for dendroclimatic studies. This is particularly crucial when reconstructing precipitation itself, as these datasets are more likely to capture the fine-scale spatial variability inherent in rainfall patterns that directly influences tree growth, thus reducing calibration uncertainty compared to lower-resolution alternatives like CRU. However, there appears to be a practical threshold for resolution benefits. For instance, the improvement gained by moving from monthly (e.g., CRU) to daily or high spatial resolution (e.g., ROCIO) was significant. Yet, while the very highest resolution dataset tested (SiCLIMA) maintained a strong signal, it did not provide a substantial *further* improvement over ROCIO in our specific study area, suggesting that beyond a certain point of high resolution, additional refinement may yield diminishing returns in terms of calibration skill. Nevertheless, the final reconstruction, calibrated with the high-resolution ROCIO data, provides an annual perspective on hydroclimatic variability over the last five centuries, offering critical long-term context for understanding recent and future climate trends in this recognized climate change hotspot (IPCC, 2021).

Situating this reconstruction within the broader context of Mediterranean paleoclimatology reveals both consistencies and important distinctions. While numerous studies have reconstructed drought indices like PDSI, SPI or SPEI across the basin, reconstructions focusing specifically on quantitative precipitation amounts are less common. Our work adds significantly to this latter category by providing a long, calibrated record for the Iberian Range. Direct comparison with index-based reconstructions must be cautious due to differences in the reconstructed variables and the specific seasonal targets. For instance, the SPI reconstruction by Tejedor et al. (2016) developed using similar *Pinus* species also in the Iberian Range, targeted a 12-month SPI ending in July. This reflects a different integration period and index calculation method compared to our focus on 320-day cumulative precipitation ending June 30th. Nonetheless, the notable overlap in identified extreme dry years (e.g., 1741, 1803, 1879, 1994 and 2012, comparing our Figure 7 extremes with their findings) strongly suggests that despite the different methodological targets, the underlying tree-ring network common to both studies reliably captures the signal of major hydroclimatic events driven by significant precipitation deficits in the region. This congruence across reconstructions focused on different aspects of hydroclimate provides mutual validation for the occurrence and timing of significant historical droughts captured by the *Pinus* chronologies in the Iberian Range. Comparison with larger-scale reconstructions like the Old World Drought Atlas (OWDA; Cook et al. (2015), highlights the distinct nature of our regional precipitation reconstruction. When our August–June precipitation series was directly correlated with the OWDA self-calibrating PDSI data for the grid point nearest our study region, the resulting overall correlation over the common period (1505–2012) was very low

(Spearman's r = 0.11). Furthermore, an analysis using 30-year moving correlations (not shown) only identified some sustained periods of significant agreement, especially since the second half of the 18th century. This weak relationship underscores the

substantial differences between a regional, quantitative precipitation reconstruction and a continental-scale drought index (PDSI) which integrates temperature and operates at a coarser resolution. It suggests that local and regional precipitation dynamics in the Iberian Range can diverge considerably from the broader patterns captured by the OWDA. Consequently, while large-scale drought atlases are invaluable for understanding continental synoptic patterns, our findings stress the critical need for developing targeted, local-to-regional precipitation reconstructions to capture the hydroclimatic variability most

relevant for specific ecosystems and water resource management in areas like the Iberian Range.

Our comparison with independent historical data, specifically rogation ceremony records compiled and processed according to established methodologies (Barriendos, 1997; Martín-Vide and Barriendos, 1995; Tejedor et al., 2019), provides valuable context. The statistically significant negative correlations observed between our precipitation reconstruction and the combined regional DI (late 18th-early 19th centuries) and the local Teruel DI (similar period) indicate a convergence between the

environmental signal captured by tree rings and the societal perception of drought reflected in these historical records. This alignment during the late 18th and early 19th centuries is particularly meaningful, as recent high-resolution analyses integrating historical and instrumental data have confirmed both the exceptional frequency and severity of droughts and wet years during this specific period and the reliability of rogation records for capturing these events prior to documented inconsistencies emerging after 1836 (Barriendos et al., 2024).

While fundamental differences exist between continuous biological proxies and curated documentary series based on societal responses (Brázdil et al., 2005), the periods of agreement suggest both are capturing hydroclimatic fluctuations severe enough to impact ecosystems and human activities. The lack of strong correlation in earlier periods or variable wet-year correspondence may stem from key distinctions between these proxies, such as the annually integrated, growth-season signal in tree rings potentially influenced by biological memory, compared to the event-triggered nature of rogation records reflecting

perceived agricultural stress at specific times of the year, as well as changes in recording practices over time. Nevertheless, the documented alignment offers compelling independent support for the reconstruction's capacity to reflect meaningful past hydroclimatic variability.

The resulting 520-year precipitation reconstruction reveals substantial hydroclimatic variability in the Iberian Range, a key mountain range within the western Mediterranean, over the past five centuries. These findings provide crucial empirical data

to assess whether recent hydroclimatic extremes, whose attribution is often debated regarding the roles of anthropogenic climate change versus internal variability (Vicente-Serrano et al., 2025), are truly unprecedented in the long-term context. While our reconstruction suggests periods of significant variability throughout the record, including clusters of extreme years in the 16th, 18th and early 19th centuries, it crucially reveals an intensification in the frequency and magnitude of both wet and dry extremes during the late 20th and early 21st centuries that appears unique within the context of the last 500 years. Our

multi-centennial perspective aligns with reports suggesting increased hydroclimatic instability in recent decades across Europe (Büntgen et al., 2021) and underscores that this recent intensification has no clear analogue over the investigated period. The establishment of such long-term baselines is essential for accurately attributing recent changes and putting them in the socio-economic context from the past centuries (Büntgen et al., 2011). The implications of this reconstruction extend to understanding ecosystem dynamics and water resource management in the western Mediterranean. The Iberian Range hosts important forest ecosystems dominated by the *Pinus* species used in this study, which are sensitive to precipitation and droughts (Camarero et al., 2013; Pasho et al., 2011a). The reconstructed history of precipitation provides a long-term ecological context, revealing the range of conditions these forests have endured.

The strength of this study lies in several key aspects. First and foremost, the exceptional length of the reconstruction (over 500 years) provides a rare and valuable long-term hydroclimatic perspective for the Iberian Range. Second, a critical strength stems from the rigorous approach to climate data evaluation and target variable identification: we assessed multiple gridded precipitation datasets (Sun et al., 2018; Tapiador et al., 2012; Xiang et al., 2021), including high-resolution products specific to Spain, and utilized daily correlation analysis (Torbenson et al., 2024) with the chosen high-resolution dataset to precisely define the optimal growth window. This approach is particularly relevant in this precipitation-limited environment and allows for a more nuanced capture of the climate signal compared to relying solely on pre-defined monthly periods. Third, the resulting reconstruction focuses on quantitative precipitation, offering direct estimates of past weather conditions rather than relying on derived drought indices. Furthermore, the clearly defined spatial representativeness (Figure 8) centred on eastern and central Iberia, provides a solid basis for the interpretation and application of the reconstruction.

Future research could build upon this work in several directions. Efforts to extend the reconstruction further back in time, potentially into the medieval period, by sampling older living trees or utilizing wood from historical structures (Esper et al., 2015) would be highly valuable. The incorporation of additional tree-ring parameters, such as latewood width (Tejedor et al., 2017), could provide complementary climatic information and potentially refine the reconstruction or allow targeting different seasons or variables. Expanding the spatial network of chronologies within the Iberian Range and adjacent regions could improve spatial detail and allow for more nuanced analysis of regional climate patterns. Investigating the large-scale atmospheric circulation patterns associated with the reconstructed wet and dry extremes could enhance mechanistic understanding of regional hydroclimate drivers. Finally, comparing this empirically derived reconstruction with output from paleoclimate model simulations could provide valuable insights into both model performance and the dynamics of past climate variability.

**Data availability**

The tree-ring chronology used in this paper will be uploaded to the International Tree-Ring Databank (ITRDB).

## Author contribution

ET and MMM conceptualized the study; MMM conducted the data analysis and wrote the original manuscript draft, with methodological development support from ET and GB. MB and GB provided and curated historical documentary resources. ET, GB, MAS, MdL, EMdC, JE and MdL provided supervision throughout the project. All authors contributed to the interpretation of the results and the critical review and editing of the final manuscript.

## Competing interests

The authors declare that they have no conflict of interest.

## Acknowledgements

We thank Iván Alfaro-Rodríguez for his valuable assistance during fieldwork. We also gratefully acknowledge José Creus Novau for his foundational sampling work in the study area.

## Financial support

ET and MMM are funded by the Comunidad de Madrid program Atracción Talento "César Nombela" grant number 2023-T1/ECO-29118. ET is further supported by the Fundación BBVA through the "Beca Leonardo de Investigación Científica y Creación Cultural", project MEDIRINGS (LEO24-1-12749-CMT-CCT-32). This research is the result of the MEDYRISK project (PID2024-160542OB-I00) funded by MICIU / AEI / 10.13039/501100011033 / FEDER, EU). This paper is a contribution from the Hydrology and Climate Change Laboratory (www.floodsresearch.com; X: floods_research; Instagram: @floods_research).

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
