# Peer review of "A five-century tree-ring record from Spain reveals recent intensification of western Mediterranean precipitation extremes"

_EGUsphere, 2025_

## Author Comment (AC1)

Dear Editor, this is the point-by-point response to Reviewer1's comments and suggestions.

We thank the reviewer for providing such thorough and constructive feedback on our manuscript. We believe the revisions we have made have substantially improved the clarity, rigor, and impact of our paper.

The referee's comments are shown in black. Our responses are shown in blue and the added or modified texts are shown in blue italics.

**Main concerns**

• Concerns 1 and 2:
1. "From the Title and Abstract, we expected to see sufficient evidences about "recent intensification of hydroclimatic extremes", but there is few in Results and Discussion about it. Most of the results are about the reconstruction development and comparisons with rogation-derived drought indices.
2. From Fig.7, we can see a little bit increase in frequency and intensity of wet and drought extremes after 2000 CE, but it is not proven. The variance of the reconstructed precipitation should be a good indicator to show it. Additionally, the running RABR should be shown in Fig. 6 to help evaluating the impacts of sample depth and inter-series correlations on variance of precipitation."

In order to address concerns 1 and 2, we have now implemented two major improvements to show sufficient evidence of recent intensification of precipitation extremes:
a) **Quantitative analysis of extremes**: we have created a new table that quantifies the frequency of extreme dry and wet years per century as follows:

*"Visual inspection of **Error! Reference source not found.** highlights periods characterized by distinct wet or dry conditions, as well as shifts in variability. For instance, the reconstruction identifies notable drought periods, with years like 1526, 1527, 1879, 1931, 2005 and 2012 falling below the 5th percentile, and exceptionally wet periods, with years such as 1534, 1546, 1575, 1645, 1716, 1940, 2010 and 2013 exceeding the 95th percentile. To further investigate the temporal dynamics of hydroclimatic variability, we calculated a 51-year running standard deviation of the reconstructed precipitation series. This metric, which quantifies the magnitude of year-to-year fluctuations (Von Storch and Swiers, 1999), is displayed in Fig. 7 (green dashed line, right-hand axis). The running SD reveals a period of pronounced climatic stability with low variability centred around the mid-19th century. This contrasts sharply with a subsequent, persistent rise in volatility that begins around the start of the 20th century and accelerates markedly after 1975, reaching values that are unprecedented in the context of our 519-year reconstruction.*

*To provide robust quantitative evidence for this observed intensification, we analyzed the frequency of extreme events per century (Table 1). The analysis reveals a high contrast between the hydroclimatically stable 19th century, which recorded only nine events outside the P10–P90 range, and the recent period. The 21st century, though spanning only 24 years, has already accumulated eight such events. This intensification is most pronounced for the rarest occurrences: the 2001–2024 period has registered four exceptionally rare events (two <P01 and two >P99). This corresponds to an occurrence rate of 16.7 %, an order of magnitude higher than the 1.6 % average rate for such events across the preceding five centuries (1505–2000). These numerical results offer strong statistical support for the visual evidence from the running SD and confirm the recent and anomalous intensification of hydroclimatic extremes in our study region."*

*Table 1: Frequency of extreme events per century, with breakdown by severity.*

| Period (CE) | Total years | Dry years (<P10) | <P05 | <P01 | Wet years (>P90) | >P95 | >P99 | Total extremes |
|---|---|---|---|---|---|---|---|---|
| 1505–1600 | 96 | 10 | 4 | 2 | 12 | 5 | 1 | 22.9 % |
| 1601–1700 | 100 | 13 | 7 | 0 | 9 | 6 | 1 | 22.0 % |
| 1701–1800 | 100 | 8 | 3 | 0 | 9 | 3 | 1 | 17.0 % |
| 1801–1900 | 100 | 6 | 3 | 1 | 3 | 1 | 0 | 9.0 % |
| 1901–2000 | 100 | 12 | 6 | 1 | 14 | 8 | 1 | 26.0 % |
| 2001–2024 | 24 | 3 | 3 | 2 | 5 | 3 | 2 | 33.3 % |

b) **Visualization of variability**: We have revised our main reconstruction figure (Figure 7) to include a 51-year running standard deviation of the reconstructed precipitation. This new curve visually demonstrates the marked increase in precipitation volatility in recent decades, corroborating the findings from our new Table 2.

[Figure]

*Figure 7: Annual rainfall reconstruction between August 16 of the previous year and June 30 of the current year back to 1505 (grey curve). A 11-year rolling mean is shown in brown. Dark blue squares represent years above 99th percentile, light blue squares between 95th and 99th percentiles, dark red below 1st percentile and light red between 1st and 5th. The variability in the reconstructed values is shown with a green dashed line showing a 51-year running standard deviation.*

We have also revised the figure showing the chronology statistics (Figure 2; we believe the referee's mention of Fig. 6 was a typo). This figure now includes a 51-year running Rbar, as suggested, to provide a clearer picture of the stability of the common signal over time, alongside SSS and sample depth. We are confident that these additions provide the quantitative evidence required to fully support our conclusions.

[Figure]

*Figure 2: Residual chronology (blue), SSS (red), running 51-year Rbar (yellow) and number of samples per year and SSS (green).*

• Concern 3: "Line 340-407, there are so many results (or discussion) about the comparisons between precipitation reconstruction and rogation-derived drought event, even including some correlations (Line 361, 366, 373,…), but no one figure and table to show these results. By using the method of giving the examples (a lot of "For instance") to show the alignment of tree-ring based precipitation reconstruction with rogation ceremony records is not sufficient to support the precipitation influence on ecosystem and society, as maybe more disagree years happened."

Fair enough. Our approach was chosen to carefully handle the fundamental differences between these two proxy types: a continuous, annually-integrated biological record (tree rings) versus a discrete, event-triggered societal response (rogations), which makes a simple year-to-year correlation plot potentially misleading. To address the reviewer's point directly and improve clarity, we have now added Table 3 after line 375, which summarizes the periods of statistically significant correlation from our moving window analysis. We acknowledge the inherent limitations of such a comparison, but the documented periods of significant alignment provide valuable historical evidence, demonstrating that our reconstruction captures hydroclimatic extremes that were severe enough to impact the socioeconomic and natural systems.

**Table 2: Main periods of significant correlation between reconstructed precipitation and regional rogation-derived drought indices. The table highlights the core intervals where a statistically significant negative correlation (Spearman r, p < 0.05) was identified between the reconstructed precipitation and the two main regional rogation clusters, based on a 50-year moving window analysis. For each regional index, the table displays the period of significance, the average Spearman coefficient across that period and a measure of data density, expressed as the number of years with recorded rogation ceremonies relative to the total duration of the interval.**

| Area | Period (CE) | Mean Spearman correlation | Number of years with rogation ceremonies within the period |
|---|---|---|---|
| Mediterranean Coast (DIMED) | 1738–1801 | –0.32 | 58/64 |
| Ebro Valley (DIEV) | 1747–1808 | –0.35 | 56/62 |

• Concern 4: " The organization of the manuscript is poor. (1) First, there are some repeat information in Discussion part. Such as, Line 416-419, "The resulting regional residual chronology demonstrates strong internal coherence (Rbar = 0.273), similar to those from other Mediterranean hydroclimatic reconstructions such as Esper et al. (2021); Klippel et al. (2018); Tejedor et al. (2016) (Rbar = 0.28; 0.31; 0.29). Our chronology retains a reliable common signal back to 1505 CE, as indicated by an SSS value consistently exceeding the 0.85 threshold (Buras, 2017; Cook and

Kairiukstis, 1990; Wigley et al., 1984)", which is repeat with Results. (2) Line 489-492, "Visual inspection of Fig. 7 highlights periods characterized by distinct wet or dry conditions, as well as shifts in variability. For instance, the reconstruction identifies notable drought periods, with years like 1526, 1527, 1879, 1931, 2005 and 2012 falling below the 5th percentile, and exceptionally wet periods, with years such as 1534, 1546, 1575, 1645, 1716, 1940, 2010 and 2013 exceeding the 95th percentile." should be represented in Results part, as actually there is no discussion about it here. (3) The paragraph of Line 478-487 should be moved to the second paragraph from bottom as a summary to highlight the key aspects of this paper. Now, it is in the middle of discussion and disturbed the discussion about reconstructed precipitation."

We have done the following:
-Removed the repeated information on chronology statistics from the Discussion (lines 416–419):
*"The resulting regional residual chronology demonstrates strong internal coherence (Rbar = 0.273), similar to those from other Mediterranean hydroclimatic reconstructions such as Esper et al. (2021); Klippel et al. (2018); Tejedor et al. (2016) (Rbar = 0.28; 0.31; 0.29). Our chronology retains a reliable common signal back to 1505 CE, as indicated by an SSS value consistently exceeding the 0.85 threshold (Buras, 2017; Cook and Kairiukstis, 1990; Wigley et al., 1984)."*

-Moved the description of specific extreme years from the Discussion to the Results section (former lines 489–492; now moved to where Figure 7 is first introduced):

*"Visual inspection of **Error! Reference source not found.** highlights periods characterized by distinct wet or dry conditions, as well as shifts in variability. For instance, the reconstruction identifies notable drought periods, with years like 1526, 1527, 1879, 1931, 2005 and 2012 falling below the 5th percentile, and exceptionally wet periods, with years such as 1534, 1546, 1575, 1645, 1716, 1940, 2010 and 2013 exceeding the 95th percentile".*

Relocated the paragraph (lines 478–487) summarizing the study's strengths to the end of the Discussion, as it provides an excellent foundation for the subsequent interpretation.

*"The strength of this study lies in several key aspects. First and foremost, the exceptional length of the reconstruction (over 500 years) provides a rare and valuable long-term hydroclimatic perspective for the Iberian Range. Second, a critical strength stems from the rigorous approach to climate data evaluation and target variable identification: we assessed multiple gridded precipitation datasets (Sun et al., 2018; Tapiador et al., 2012; Xiang et al., 2021), including high-resolution products specific to Spain, and utilized daily correlation analysis (Torbenson et al., 2024) with the chosen high-resolution dataset to precisely define the optimal growth window. This approach is particularly relevant in this precipitation-limited environment and allows for a more nuanced capture of the climate signal compared to relying solely on pre-defined monthly periods. Third, the resulting reconstruction focuses on quantitative precipitation, offering direct estimates of past weather conditions rather than relying on derived drought indices. Furthermore, the clearly defined spatial representativeness (**Error! Reference source not found.**) centred on eastern and central Iberia, provides a solid basis for the interpretation and application of the reconstruction."*

**Response to minor problems**
1. About the Title, "precipitation extremes" is more exact than "hydroclimatic extremes" to highlight the study gap.

As both reviewers have suggested to replace hydroclimate with precipitation, we have modified the title to read: *"A five-century tree-ring record from Spain reveals recent intensification of western Mediterranean precipitation extremes"*

2. Line 89, "June is the month with the highest pluviosity, followed by May" is inconsistent with Fig.1B, which shown precipitation in May is the highest, followed by April.

Thank you for spotting this error. Corrected to: *"May is the month with the highest pluviosity, followed by April"*

3. Tree-ring series from five sites and two species were used for developing one chronology. How about the correlations between sites and species, and the uniformity of five chronologies at high frequency and low frequency variability? These informations could be plotted in Supplementary materials.

As suggested, we have now added two new tables in the Supplementary materials showing the correlations between the five individual site chronologies to demonstrate their coherence. Also, there's a new paragraph in the main text, at the end of the 2.2 (Tree chronology development) subsection.

*"To justify the creation of a regional chronology, inter-site correlations were calculated on the standardized residual series (see Supplementary Table S1A and S1B). The analysis shows positive correlations across the entire network during the common instrumental overlapping period, with the all the sites being statistically significant ($p < 0.05$). The correlations for the full interseries overlapping period also show high correlations between sites, except between Javalambre and Valdecuenca sites ($r = 0.19$), which does not reach the 0.05 significance level, the overall strong coherence, with 9 out of 10 inter-site correlations being highly significant, indicates a robust common climatic signal across the network and validates their combination into a regional composite. These correlations are based on robust overlapping periods between sites, ranging from 78 to 520 years (Table S2), with Javalambre and Valdecuenca overlapping only 97 years."*

**Table S3: Intersite Pearson correlation matrices for the five residual site chronologies. Two different time periods are presented to assess the temporal coherence of the network. A) Common instrumental period (1952–1993): correlations are calculated for each site pair using all of their available common years. B) Full individual overlapping periods: correlations are calculated for all sites over a single, consistent period. All correlations are statistically significant ($p < 0.05$), with the exception of the JAR-VAN pair in the full overlapping period analysis.**

| A. Common overlapping period during the calibration period (1952-1993) | | | | | |
| --- | --- | --- | --- | --- | --- |
| | BEL | JAR | LIN | MOS | VAN |
| BEL | 1.00 | 0.75 | 0.66 | 0.6 | 0.6 |
| JAR | 0.75 | 1.00 | 0.55 | 0.44 | 0.47 |
| LIN | 0.66 | 0.55 | 1.00 | 0.55 | 0.70 |
| MOS | 0.60 | 0.44 | 0.55 | 1.00 | 0.79 |
| VAN | 0.60 | 0.47 | 0.70 | 0.79 | 1.00 |

| B. Individual overlapping periods | | | | | |
| --- | --- | --- | --- | --- | --- |
| | BEL | JAR | LIN | MOS | VAN |
| BEL | 1.00 | 0.38 | 0.50 | 0.52 | 0.33 |
| JAR | 0.38 | 1.00 | 0.33 | 0.33 | 0.19 |
| LIN | 0.50 | 0.33 | 1.00 | 0.38 | 0.32 |
| MOS | 0.52 | 0.33 | 0.38 | 1.00 | 0.40 |
| VAN | 0.33 | 0.19 | 0.32 | 0.40 | 1.00 |

**Table S4: Overlapping periods between site chronologies. Number of overlapping years between each pair of site chronologies used for the correlation analysis in Table S1. The diagonal (in bold) shows the total length of each individual site chronology.**

| | BEL | JAR | LIN | MOS | VAN |
| --- | --- | --- | --- | --- | --- |
| BEL | **581** | 520 | 297 | 314 | 109 |
| JAR | 520 | **520** | 297 | 302 | 97 |
| LIN | 297 | 297 | **297** | 283 | 78 |
| MOS | 314 | 302 | 283 | **314** | 109 |
| VAN | 109 | 97 | 78 | 109 | **109** |

4. Line 412-413, "capture the critical moisture accumulation phase influencing annual growth (late summer, autumn, winter and spring/early summer)," how to understand the autumn and winter precipitation influence on tree radial growth considering trees dormancy in winter.

We have added a sentence to clarify the physiological mechanism, explaining that autumn and winter precipitation is crucial for recharging soil moisture, which supports earlywood growth at the start of the following season.

*"Physiologically, this window was strategically selected through detailed correlation analysis to capture the critical moisture accumulation phase influencing annual growth. This extended period is crucial because*

*autumn and winter precipitation, often accumulating as snowpack at these mountain sites, recharges soil moisture during tree dormancy. The release of this water during spring snowmelt then directly supports earlywood formation (Pasho et al., 2011a, b), while the window still excludes the peak summer drought period (July to mid-August) where growth is typically limited by intense water stress (Camarero et al., 2013).”*

5. Line 445, “comparing our Fig. 6 extremes with their findings”, should be Fig. 7, right?

This was a typo and it is *“Fig. 7”.*

6. Line 472-474, “The lack of strong correlation in earlier periods… influenced by biological memory, …” seems inconsistent with the feature of RES chronology with “pre-whitening to reduce autocorrelation”.

The paragraph has now been rephrased to:

*“The lack of strong correlation in earlier periods or variable wet-year correspondence may stem from several factors. Firstly, and most importantly, while our chronology meets the SSS > 0.85 threshold for reliability throughout, the inherent decrease in sample replication in the early part of the record leads to a statistically weaker common signal compared to the well-replicated recent period (Buras, 2017; Wigley et al., 1984). This reduced signal strength naturally makes robust correlations with any external proxy, including rogation records, more challenging to achieve. Secondly, fundamental distinctions between the proxies remain: our reconstruction represents an annually integrated, 320-day signal, whereas rogation records are event-triggered responses to perceived agricultural and social stress, which may not always align seasonally.”*

7. 5 is not clear and takes up too much space.

We appreciate the feedback on this figure. While we acknowledge it is dense, we believe it is crucial for demonstrating the rigour of our data-driven approach to defining the optimal climate window, which is a key strength of the study, since we are using daily precipitation data. We have refined the figure caption to better guide the reader in its interpretation and we have also considered reducing the period shown (and the window length shown) to make it more compact.

[Figure]

*Figure 5: Daily climate-growth response analysis showing Pearson correlation coefficients between the residual tree-ring chronology and daily accumulated precipitation. The heatmap explores a vast parameter space to identify the optimal climate window influencing tree growth. The y-axis represents the width of the precipitation window in days (from 200 to 450 days). The x-axis represents the ending day of this window, spanning from January to December of the current growth year.*

*The colour of each pixel indicates the strength of the correlation (r). Only statistically significant correlations (p < 0.05) are shown. The yellow box highlights the pixel with the maximum positive correlation (r = 0.749), which corresponds to a window of 324 days ending on July 1st.*

---

## Author Comment (AC3)

Dear Editor, this is the point-by-point response to Reviewer2's comments and suggestions.

We thank the reviewer for their constructive feedback. We believe the manuscript is significantly improved as a result of these revisions.

The referee's comments are shown in black. Our responses are shown in blue and the added or modified texts are shown in blue italics.

**Major comments**

1. Usually it is a good idea in these cases to implement "variance stabilization" in chronology development: adjust the time-varying chronology variance for the changing sample size (number of trees or cores). If variance stabilization was used, you should report that in the methods. If not, I suggest doing a quick check to see if it makes a difference to conclusions.

   To address this concern, we have performed a detailed comparative analysis between the reconstruction used in our original manuscript (based on a residual chronology) and an alternative reconstruction. This alternative was developed using a variance stabilization method, which corrects for artifactual changes in variance that arise from fluctuating sample replication over time (implemented as the `chron.stabilized` function in the dplR package).

   The comparative analysis was approached from two perspectives:

   1. **Comparison of the full time series:** The two reconstructions are overwhelmingly similar, exhibiting a Pearson correlation of **r = 0.92**, which indicates they share approximately 85 % of their variance. A paired t-test found no significant difference in their means (p = 0.087), and a Kolmogorov-Smirnov test confirmed that their overall probability distributions are statistically indistinguishable (p = 0.246).

   2. **Analysis of the frequency and distribution of extremes:** Since a key part of our study's conclusions relies on extreme events, we performed specific statistical tests (McNemar's and Chi-squared tests) to compare the identification of these events. The results were conclusive: **no statistically significant differences were found** either in the classification of extreme years (McNemar's tests, p = 1.0 in all cases) or in their temporal distribution across centuries (Chi-squared tests, p > 0.48 in all cases).

   Beyond this empirical validation, we also find the reviewer's point insightful and have addressed it with an additional, direct test. The concern is particularly relevant to our chronology, where the combination of historical and modern sampling campaigns results in a stepwise decrease in sample replication in the most recent decades. To specifically test the influence of this structure, we created an independent chronology using only the modern, high-replication samples, thus removing the drop in sample depth. This "modern-only" reconstruction is statistically indistinguishable from our main chronology, confirming that our results are not an artifact of changing replication. This provides direct, data-driven evidence that the main conclusion of our study is robust and not biased by the chronology's structure.

   Taken together, these analyses robustly demonstrate that the main conclusion of our study is not an artifact of the chosen chronology development method. The statistical similarity between both reconstructions, particularly with respect to extreme events, further confirms the validity of our findings.

   We have updated the **Chronology development** subsection of the manuscript (3.1, after current line 227) to include a summary of this sensitivity analysis:

   *"Additionally, to ensure the robustness of our findings to the chosen methodology, a comprehensive*

*sensitivity analysis was performed. An alternative reconstruction was developed using a variance stabilization method (chron.stabilized in dplR; Wigley et al., 1984; Frank et al., 2007), which corrects for variance changes associated with fluctuating sample replication over time. A thorough statistical comparison revealed that both reconstructions are highly similar (Pearson r = 0.92) and exhibit no significant differences in their means (paired t-test, p = 0.087) or their overall probability distributions (Kolmogorov-Smirnov test, p = 0.246). Crucially, a specific analysis of extreme events showed no statistically significant differences in their frequency or temporal distribution (McNemar's and Chi-squared tests, p > 0.48). Given that the study's conclusions are robust to the choice of method, and considering the theoretical suitability of the residual chronology for calibration, the latter was retained for all analyses presented. Full details of this comparative analysis can be found in the Supplementary material (Table S3)."*

Added in the Supplementary material:

*Table S1: Statistical comparison of the full time series from the residual and variance-stabilized reconstructions over the common period 1505-2024.*

| Test | Statistic | Value | p-value | Interpretation |
|---|---|---|---|---|
| *Pearsons correlation* | *r* | *0.920* | *< 0.001* | *Very strong positive correlation between the series.* |
| *Paired t-test* | *t (df = 519)* | *1.715* | *0.087* | *No significant difference between the mean values of the two series.* |
| *Kolmogorov-Smirnov* | *D* | *0.063* | *0.246* | *The probability distributions of the two series are statistically indistinguishable.* |

2.  Distinction of "hydroclimate" from "precipitation." Water availability, which is mentioned in several places in the paper, is a function of not just precipitation but also of evapotranspiration. The authors claim that a strong point of this paper compared with previous works is that it addresses precipitation rather than PDSI or some other type of drought index. This may be a strong point in terms of synoptic climatology and delivery of precipitation to the region, but not necessarily in terms of water resources availability or ecosystem stress. It seems likely that the tree growth of these drought sensitive species would not better reflect the combined stress of low precipitation and high temperature than the stress of low precipitation alone. A natural question is whether the impressive calibration strength (R-squared) for the reconstruction model might not be even more impressive for some sort of drought index that incorporating influence of evapotranspiration. I think the paper could benefit from some comparison statistics. If checking against and index such as SPEI or PDSI, the standard rather than the residual chronology might actually be worth looking at because the strong temperature trend associated with regional warming is a low-frequency signal that could have been removed by autoregressive modeling during standardization. Such additional analysis, if done, would mainly be a discussion point, as readers may wonder why the calibration of a drought-sensitive chronology is not stronger with SPEI, say, than with precipitation.

and

3.  I appreciate the rigorous daily climate analysis, which resulted in identification of an optimum window, but wonder about the sampling variability and whether you are making too much of the difference in a annual window and a 320-day window. In the minor comments I bring this up again and suggest adding maybe a sentence about this issue in the discussion. Almost certainly there is no significant different in correlation for the selected 320-day window and the highest 365-day (annual) window.

Regarding major comments 2 and 3, we have reviewed the text and changed "water availability" as it follows:

-Lines 77 and 485: replaced with "weather conditions".

-Line 504: replaced with "precipitation and droughts".

We have checked the correlation against SPEI, and added the following (after current line 419):

*"A relevant question concerns the choice of the reconstruction target, specifically why precipitation was selected over a more integrated drought index like the Standardized Precipitation-Evapotranspiration Index (SPEI) (Beguería and Vicente-Serrano, 2011). To address this, we conducted a direct comparative analysis to empirically determine the optimal target variable. We performed a series of calibrations testing both our residual and standard chronologies against three potential climate targets: our 320-day precipitation window, a standard 12-month precipitation window, and a 12-month SPEI.*

*The results, detailed in the Supplementary material (Table S4), demonstrate a clear and consistent hierarchy in model performance. Our selected 320-day window produced a stronger calibration than a standard 12-month window. Crucially, the model with the highest predictive skill was the one used in our study, which calibrates the residual chronology against 320-day precipitation Therefore, the selection of this specific target and predictor is not only based on our initial response function analysis but is also validated by this comprehensive comparative test as the statistically most robust approach for this dataset. It is worth noting, however, that while the 320-day window is empirically superior, the calibration strength of the 12-month window remains high, with only minor differences between the two."*

Added in the Supplementary material:

*Table S2: Comparison of calibration statistics for different climate targets and chronology types. The climate window is 320 days ending June 30 for precipitation and 12 months ending June for SPEI and 12-month precipitation. All calibrations are for the period 1951–2022.*

| Chronology type | Climate target | Calibration Pearson r | p-value |
|---|---|---|---|
| *Residual* | *Precipitation (320 days, ending June 30)* | *0.749* | *<0.001* |
| *Standard* | *Precipitation (320 days, ending June 30)* | *0.746* | *<0.001* |
| *Standard* | *Precipitation (365 days, ending June 30)* | *0.741* | *<0.001* |
| *Residual* | *Precipitation (365 days, ending June 30)* | *0.738* | *<0.001* |
| *Standard* | *SPEI (12-month scale, ending June)* | *0.723* | *<0.001* |
| *Residual* | *SPEI (12-month scale, ending June)* | *0.721* | *<0.001* |

**Minor comments**

1. Title: Consider substituting "precipitation" for "hydroclimate," in the title, because precipitation is what has been reconstructed. Precipitation is just one aspect of hydroclimate. Runoff and streamflow are the sum of net precipitation (P-ET) and are what I think of as key components of hydroclimate, though this is an arguable distinction.

   Done. New title: "*A five-century tree-ring record from Spain reveals recent intensification of western Mediterranean precipitation extremes*".

2. L77. I disagree that precipitation provides a "more direct measure of past water availability than some drought index. Net precipitation (P-ET) is one possible drought index, and is actually more relevant to water availability the precipitation alone. Of course, precipitation is more directly link weather delivery systems that P-ET, which depends on vegetation and other land surface factors.

   Fair enough.
   -Lines 77 and 485: replaced with "*weather conditions*".

-Line 504: replaced with *"precipitation and droughts"*.

3. L90. "pluviosity" is an overblown word when used here for "precipitation, " which explicitly is what is shown in the climate diagram, and what is measured in a rain gauge.

Done, now changed to *"precipitation"*.

4. L90. On the climate diagram I see May followed by April, not June followed by May, as the months of highest precipitation.

Thank you. Now corrected to*: "May is the month with the highest precipitation, followed by April"*.

5. L92. Looks to me like Feb is a drier month month than Aug. The statement about July and Aug being driest month applies only if just considering summer .

Ok. Now changed to: *"... showing July and August as the months with the highest mean temperatures, with July also being the driest month"*.

6. L114. "Campaigns at…".

changed "in" to "at".

7. L124. Standardization description needs a bit more information. Was the ratio or difference method used for converting ring widths to indices?  Was the site chronology computed as an arithmetic mean or biweight mean of core indices?  Was ariance stabilization applied to adjust variance changes in site chronology to time-varying sample size (see Major comments)? Did you compute both standard and residual versions of the chronology, and why did you select the residual version for the reconstruction

and

8. L 128. How many trees are represented by the 173 series? I'm assuming probably more than one core per sampled tree.

Minor comments 7 and 8 have been taken into account in order to improve the clarity of the explanations provided regarding the chronology development. In addition to the response to major comment 1, which also covers these topics, current lines 124–129 have now been changed to:

*"After measuring the samples, each individual ring-width series was standardized to remove age/size-related trends and to minimize non-climatic noise. This was performed using the dplR package in R. Following an adaptive approach based on series length, a negative exponential curve or a cubic smoothing spline of variable stiffness was fitted to each series. Tree-ring indices were then calculated as ratios by dividing the raw ring-width measurements by the fitted curve values.*

*These individual indices were then combined into a regional chronology using a biweight robust mean, a method that minimizes the influence of outliers. Both a standard and a residual version of the chronology were computed. The residual chronology was selected for the final reconstruction because it is generated via prewhitening, a procedure that fits an autoregressive (AR) model to the standard chronology and removes the statistical autocorrelation inherent in tree growth. (Cook and Kairiukstis, 1990).*

*In addition, a sensitivity analysis was conducted to test the effect of an alternative method, variance stabilization, on our results. As detailed in the Supplementary Table S.3, this approach did not improve the calibration skill. Therefore, the residual chronology was retained as it worked as the strongest predictor. After excluding a subset of older series with low signal strength (correlation < 0.3) or that were inaccessible for remeasurement, the final dataset consisted of 173 individual tree-ring series from 103 different trees."*

9. L 153. " with12-" ….insert a space

Done.

10. L159. "robust" -- I assume the daily window selected is robust to selected segment of the climate-chronology overlap (e.g., approximately same day window if analysis repeated on separate halves of the record)

Yes, we tested in separate halves of the record (1952–1986 and 1987–2022) and both the windows and ending dates of the windows were similar in the two halves.

11. L 163. "linear transfer function model" --- the statistical reconstruction method could be described more directly as "simple linear regression of the target predictand on the site chronology."

Ok. Changed to *"simple linear regression of the target predictand on the site chronology"*.

12. L208 "Such a low AR1 value indicates that the standardization effectively removed most of the tree-ring memory persistence inherent in tree growth, yielding a time series suitable for robust correlation analysis with external environmental variables, such as climate." Yes, as long as the target predictand also has no autocorrelation. Is that so for precipitation in this region? Also, in regression, analysis of residuals check usually included first order autocorrelation of regression residuals (e.g., by Durbin Watson statistic). It is assumed in regression that there is no significant lag-1 autocorrelation in the regression residuals.

Thanks. We ran the tests and we have edited the current lines 208–210 in the manuscript, which now say:
*"The first-order autocorrelation (AR1) coefficient for this residual chronology was 0.039. This low value indicates that standardization effectively removed the non-climatic, biological persistence inherent in tree growth. The instrumental precipitation target also exhibited negligible first-order autocorrelation (AR1 = –0.037). This ensures that both predictor and predictand are suitable for correlation analysis. More critically, the residuals of the final linear reconstruction model were tested for autocorrelation using the Durbin-Watson statistic; the test was not significant (DW = 2.14; p = 0.729), confirming that the model meets the assumption of independent errors required for linear regression (Cook and Kairiukstis, 1990)"*

13. *Fig 1 caption. "grid cells" – would help the interpretation to indicate the resolution of grid for the precipitation. .*

Modified as it follows:
*"Climate diagram showing monthly precipitation from the ROCIO 5 × 5 km grid (Peral García et al., 2017) and temperature from the Spain02 20 × 20 km grid (Herrera et al., 2016), both calculated as the average value of the 16 grid cells closest to the coordinate 40.3° N, 1.4° W, for the period 1986–2015."*

14. Fig 2 caption. Specify that the "number of samples" is cores or trees.

Ok. Changed to *"number of cores"*.

15. Fig 3. For consistency, give the grid resolution for all of the datasets in the labels along y axis.

Ok. Done.

[Figure]

Common period: 1958–2012 | * indicates p ≥ 0.05

16. Fig 4 caption. The "red dashed line" is not the residual chronology, but the reconstruction based on it.

Changed to:

*"Figure 4: Calibration of the pine residual chronology against CRU TS and ROCIO previous-year August to current-year June precipitation sums (blue curves) from 1958–2012. The red dashed line represents the chronology-based precipitation reconstruction. Grey shades represent ±1 RMSE."*

17. Fig 5 caption. There seems to be a wide range of day windows with high correlation, or with dark red shading. How much lower is the correlation for the "best" annual (365-day) period that the correlation for selected 320-day window (r=0.749)? Could this just be a result of sampling variability? Perhaps you can add a sentence or two on this in the discussion. In a related question I wondered whether the same 320 day window is identified if use different sub-periods (e.g., first and last halves) of the record.

We have responded to this as a reply to major comments 2 and 3.

---

## Referee Report (RR1)

Second review of egusphere-2025-2530 | Journal relation: CP "A five-century tree-ring record from Spain reveals recent intensification of western Mediterranean hydroclimatic extremes" Marcos Marín-Martín, Ernesto Tejedor, Gerardo Benito, Miguel A. Saz, Mariano Barriendos, Edurne Martínez del Castillo, Jan Esper, and Martín de Luis

\_\_\_\_\_

**Overview**

This paper describes a regression-based 500+ year reconstruction of seasonal-total precipitation for the Iberian Range of eastern Spain from tree-ring data of two *Pinus* tree species and five sites collected (recently updated) by the authors. The reconstruction stands out compared with previous reconstructions of dendroclimatic variables for its great length, strength of calibration signal, and calibration with precipitation rather a drought index. A key finding is recent intensification of hydroclimatic extremes consistent with climate change projections. The reconstruction is touted as a baseline for evaluating ecosystem resilience and water resource vulnerability.

The authors have satisfactorily addressed the comments in my first review. The authors went back and repeated analysis using variance-stabilized chronologies to exclude the possibility that the recent amplified extremes are an artifact of changing sample size in the tree-ring chronology. The authors also conducted additional analysis to demonstrate that signal in the tree-rings is indeed stronger in precipitation than in alternative predictand variables (e.g., SPEI) that measure a combined precipitation-evaporation signal. The additional supplemental material effectively lets the reader see that the identified seasonal daily-window response is indeed stronger than a simpler annual response, though the difference may not be significant. It is impressive that the same aggregated daily window is identified in split-sample calibration.

Minor comments in my first review were all effectively addressed.

I conclude that the paper is overall a worthy and strong contribution.